# A national scale redox clustering for quantifying $CO_2$ emissions from groundwater denitrification

Hyojin Kim[1]*, Julian Koch[2], Birgitte Hansen[1] and Rasmus Jakobsen[1]

[1]Department of Geochemistry, Geological Survey of Denmark and Greenland, Øster Voldgade 10. 1350 Copenhagen, Denmark.
[2]Department of Hydrology, Geological Survey of Denmark and Greenland, Øster Voldgade 10. 1350 Copenhagen, Denmark

* Corresponding author: Hyojin Kim (hk@geus.dk)

## Abstract

Nitrate pollution from agriculture poses a significant global threat to the environment and to public health. In groundwater, nitrate can be reduced through denitrification, a process that produces dissolved inorganic carbon (DIC) via organic carbon (OC) mineralization and/or carbonate dissolution. This DIC acts as a net anthropogenic source of atmospheric $CO_2$; however, its overall climatic impact remains poorly constrained. In this study, we quantified $CO_2$ production from groundwater denitrification across Denmark, using extensive observational datasets and national-scale modeling tools. A set of machine learning techniques was applied to cluster groundwater redox conditions and map the dominant electron donors for denitrification at the national scale. At the redox interface, denitrification was predicted to be mediated by pyrite oxidation in approximately 76% of Denmark with the remainder dominated by OC oxidation. Our results underscore the central role of hydrogeology in controlling the distribution of dominant electron donors. Spatial variability in $CO_2$ production from groundwater denitrification was governed by nitrogen leaching and prevailing denitrification pathways. Assuming complete denitrification, we estimated that groundwater denitrification produces approximately 204 kt of $CO_2$-eq. $yr^{-1}$ as DIC, of which ~50% is likely released to the atmosphere. The Intergovernmental Panel on Climate Change (IPCC) guidelines account for agricultural $CO_2$ emissions from liming, urea, and other carbon-containing fertilizers, estimated at 250, 1 and 4 kt of $CO_2$-eq. $yr^{-1}$, respectively, for Denmark in 2020. Although $CO_2$ comprises a minor share (~2%) of total agricultural GHG emissions, our findings suggest that denitrification-derived $CO_2$ should be included in agricultural GHG inventories.

## 1. Introduction

The application of nitrogen (N) inorganic fertilizers and animal manure in agriculture is essential for global food production. However, their use poses significant environmental and public health risks (Diaz and Rosenberg, 2008; Galloway et al., 2003; Howarth et al., 2011; Jensen et al., 2023; Schullehner et al., 2018; Temkin et al., 2019; Trends et al., 2008; Ward et al., 2018). Reactive N, primarily in the form of nitrate ($NO_3^-$), leaches from agricultural soils into nearby aquatic ecosystems, leading to

eutrophication and hypoxic "dead zones" in coastal areas (Diaz and Rosenberg, 2008; Galloway et al., 2003; Howarth et al., 2011). Elevated nitrate concentrations in drinking water are associated with several health risks, such as infant methemoglobinemia (blue baby syndrome). The current regulatory limit for nitrate in drinking water (50 mg/L as nitrate) is set to prevent this condition, yet recent studies have shown that the risks of certain cancers and birth defects may increase at

concentrations below this threshold (Jensen et al., 2023; Schullehner et al., 2018; Temkin et al., 2019; Ward et al., 2018).

Nitrogen fertilizers and manure are also a major source of anthropogenic greenhouse gases (GHGs; Gao and Cabrera Serrenho, 2023; Menegat et al., 2022; Mosier et al., 1998; Zamanian et al., 2018). Nitrate undergoes denitrification, a sequence of redox reactions that converts it to nitrite ($NO_2^-$), nitrogen monoxide (NO), nitrous oxide ($N_2O$) and finally inert nitrogen ($N_2$) gas. Nitrous oxide is a potent GHG, with 265 times the warming potential of carbon dioxide ($CO_2$). Nearly all global $N_2O$ emissions

originate from agricultural fertilizer applications (Ritchie et al., 2023), primarily from soils, while 10-15 % occurs via indirect pathways such as groundwater, streams, and estuaries (Gao and Cabrera Serrenho, 2023; Nielsen et al., 2022).

The use of N-fertilizers and manure also contributes to soil acidification through microbial oxidation of ammoniacal fertilizer or urea application (Barak et al., 1997):

(nitrification of ammonia): $NH_3 + 2O_2 = H^+ + NO_3^- + H_2O$

(nitrification of ammonium): $NH_4NO_3 + 2O_2 = 2H^+ + 2NO_3^- + H_2O$

(hydrolysis of urea and nitrification of products): $CO(NH_2)_2 + 4O_2 = 2H^+ + 2NO_3^- + CO_2$

Liming is commonly used to counteract acidification, but both liming and urea applications lead to $CO_2$ emissions (Zamanian

et al., 2018). In terms of $CO_2$ equivalents ($CO_2$-eq.), liming is comparable to the indirect $N_2O$ emissions (Gao and Cabrera Serrenho, 2023; Zamanian et al., 2018) while $CO_2$ from urea is relatively insignificant (Nielsen et al., 2022).

During denitrification, $CO_2$ is produced. When oxygen is completely depleted and reduced materials (electron donors) such as organic matter are available, nitrate can be reduced to $N_2$ gas, simultaneously increasing dissolved inorganic carbon (DIC) in water (Equation 1 in Table 1; Appelo and Postma, 2005; Seitzinger et al., 2006). Assuming complete denitrification, this

process mineralizes organic carbon (shown as $CH_2O$), releasing 5 moles of DIC per 4 moles of nitrate reduced. If denitrification is incomplete and terminates at $N_2O$, only 4 moles of DIC are produced per 4 moles of nitrate reduced. Pyrite can be oxidized by oxygen (Equation 3 in Table 1) as well as nitrate (Equation 2 in Table 1; Postma et al., 2012; Torrentó et al., 2010; Zhang et al., 2009). Both reactions (2 and 3) produce protons that can dissolve carbonate minerals if present (Equation 5 in Table 1).

**Table 1. Reactions considered in this study and key stoichiometric ratios of the reactions.**

| | Groundwater chemistry | | | CO$_2$ emissions pr. mole of NO$_3^-$ reduced $(\frac{+\Delta CO_2}{-\Delta NO_3^-})$** |
| --- | --- | --- | --- | --- |
| | $\frac{HCO_3^-}{Ca^{2+}+Mg^{2+}}$ | $\frac{SO_4^{2-}}{Ca^{2+}+Mg^{2+}}$ | $\frac{+\Delta DIC}{-\Delta NO_3^-}$ | |
| **Denitrification reactions** | | | | |
| (1) Denitrification with organic C oxidation: $5CH_2O + 4NO_3^- \rightarrow 2N_2 + 4HCO_3^- + H_2CO_3 + 2H_2O$ | >2 | - | 1.25 | 0.625 |
| (2) Complete denitrification with pyrite oxidation  2a) $5FeS_2 + 14NO_3^- + 4H^+ \rightarrow 7N_2 + 5Fe^{2+} + 10SO_4^{2-} + 2H_2O$  2b) $5Fe^{2+} + NO_3^- + 12H_2O \rightarrow 0.5N_2 + 5Fe(OH)_3 + 9H^+$  $= 5FeS_2 + 15NO_3^- + 10H_2O \rightarrow 7.5N_2 + 5Fe(OH)_3 + 10SO_4^{2-} + 5H^+$ | 1* | 2* | 0.33* | 0.17* |
| **Other reactions** | | | | |
| (3) Pyrite oxidation with oxygen  $FeS_2 + \frac{15}{4}O_2 + \frac{7}{2}H_2O \rightarrow Fe(OH)_3 + 2SO_4^{2-} + 4H^+$ | 1* | 0.5* | - | - |
| (4) Reversible reaction of carbonate dissolution with CO$_2$ and carbonate precipitation  $xCa^{2+} + (1-x)Mg^{2+} + 2HCO_3^-$  $\leftrightarrow Ca_xMg_{(1-x)}CO_3(s) + CO_2(g) + H_2O$ | 2 | - | - | - |
| (5) Carbonate dissolution with strong acids  $5Ca_xMg_{(1-x)}CO_3 + 5H^+ \rightarrow 5xCa^{2+} + 5(1-x)Mg^{2+} + 5HCO_3^-$ | | | | |

*Coupled with carbonate dissolution; ** In case of calcite saturation

These processes elevate DIC levels in streams and groundwater, which naturally have significantly higher partial pressure of CO$_2$ (pCO$_2$) than the atmosphere, largely due to soil respiration, resulting in CO$_2$ outgassing from these waters (Duvert et al., 2018; Macpherson, 2009; Martinsen et al., 2024). If groundwater or stream water reaches the saturation point of calcite as CO$_2$ degasses, calcite will precipitate (Equation 4 in Table 1). In this case, while one mole of DIC is re-stored as calcite, the other is released as CO$_2$. This series of processes i.e., from denitrification to calcite precipitation does not involve atmospheric CO$_2$

but is triggered by anthropogenic nitrogen input primarily from agriculture, making them a net anthropogenic source of atmospheric $CO_2$.

Globally, approximately 50 teragrams (Tg; $10^{12}$ g) of reactive N are lost from agricultural soils annually through leaching and erosion (Liu et al., 2010), and the subsurface serves as a large reservoir of reactive N (Ascott et al., 2017). Denitrification of this leached reactive N below the soil layer could represent a significant indirect pathway for agriculturally derived $CO_2$ emissions. Unlike nitrate in streams and/or estuaries which could promote primary productivity, thus fixing $CO_2$; nitrate in groundwater is simply a pollutant and a potential source of $CO_2$ when it is reduced. Furthermore compared to direct emissions from soil, which occur promptly, indirect pathways via groundwater exhibit dispersed and delayed signals due to long and variable transit times (Basu et al., 2022; Meals et al., 2010). This delay could potentially undermine the effectiveness of climate mitigation efforts and increase uncertainty of GHG inventories and future projections.

Despite its potential relevance, $CO_2$ emissions from denitrification, particularly in groundwater, have not been systematically quantified. According to the Intergovernmental Panel on Climate Change (IPCC) guidelines, anthropogenic GHG emissions from managed soils—*lands where human interventions and practices have been applied to perform production, ecological or social functions* (IPCC, 2006)—include 1) direct (i.e., soil) and indirect (i.e., stream, groundwater, and estuary) $N_2O$ emissions from N fertilizer inputs; and 2) direct (i.e., soil) $CO_2$ emissions from liming, urea and other carbon-containing fertilizers. $CO_2$ emissions from denitrification are not currently included in the IPCC framework (IPCC, 2006). Compared to methane ($CH_4$) and $N_2O$, $CO_2$ contributes a minor share of the total GHG emissions from agriculture. However, the IPCC guidelines require individual accounting for each GHG unless there are specific methodological reasons for aggregation (IPCC, 2006). Thus, all anthropogenic sources of $CO_2$ in agriculture are required to be accounted for, regardless of magnitude.

This study, therefore, aims to quantify $CO_2$ release from denitrification of nitrate derived from agricultural N fertilizer use, in the context of national GHG inventories. To enable this quantification, we assumed complete denitrification. Incomplete denitrification, which produces $N_2O$, is highly heterogeneous in space and time (Clough et al., 2007; Jahangir et al., 2013; Jurado et al., 2017; McAleer et al., 2017). In addition, $N_2O$ produced in groundwater is likely converted to $N_2$, particularly in anoxic groundwater (Jurado et al., 2017). Therefore, we concluded that assuming complete denitrification is a reasonable approximation for large-scale assessments such as this study. In Denmark, a national groundwater monitoring program provides extensive long-term groundwater chemistry data across the country (Thorling et al., 2024). Additionally, the National Nitrogen Model (den Nationale Kvælstof Model; NKM) provides a national N budget at the catchment scale (Højberg et al., 2021). These data and tools effectively transform Denmark into a virtual laboratory for quantitative and systematic investigations of $CO_2$ emissions from denitrification at the national scale. Using these resources, along with machine learning techniques, we addressed three specific objectives: 1) characterization of geochemical architecture of Danish groundwater, focusing on redox conditions and dominant electron donors for denitrification; 2) prediction of a national map of denitrification electron donors (i.e., redox cluster map); and 3) quantification of the $CO_2$ emissions from groundwater denitrification in the context of the agricultural GHG emissions in Denmark.

## 2. Method and materials

### 2.1. Groundwater chemistry data

Groundwater chemistry data were retrieved from the National Borehole Database, Jupiter ([www.geus.dk](www.geus.dk)) in November 2022. The dataset includes all the groundwater chemistry data deposited in Jupiter in the period 1890-2022. A total of 186,887 records from 36,216 screens across the country were extracted. Some wells have multiple screens. To ensure the data quality and integrity, the dataset was cleaned using five exclusion criteria: 1) wells within a 100 m buffer zone of landfill sites; 2) wells identified as contaminated with micro-organic pollutants by the Danish Environmental Protection Agency; 3) records with incomplete geographical information (x, y, and screen depth); 4) duplicate entries; and 5) records with detection limits exceeding 1 mg/L for $NO_3^-$, $SO_4^{2-}$, $Ca^{2+}$, $Mg^{2+}$, and $HCO_3^-$, indicating low analytical quality. Post-cleaning, the dataset contained 115,276 records from 24,323 screens, primarily collected between 1990-2020.

Approximately 70% of these screens were sampled only once, with varying combinations of solutes measured. For further analysis, we selected screens that 1) had at least five measurements of $NO_3^-$, $SO_4^{2-}$, $HCO_3^-$, $Ca^{2+}$, and $Mg^{2+}$ over the entire period; and 2) had at least one measurement for pH, $Fe^{2+}$, $Mn^{2+}$, $CH_4$, $NH_4^+$, $Na^+$, and $Cl^-$. This process resulted in 6,272 screens being included in the final dataset. Finally, values below detection limits were converted to half of the detection limit.

### 2.2. Characterization of redox architecture of the Danish groundwater

The cleaned dataset was analyzed to categorize redox conditions and to identify dominant processes by combining two machine learning techniques: Non-negative Matrix Factorization (NMF) and K-means clustering. The analyses were done in MATLAB using built-in functions. Both methods are widely used for identifying sources and underlying processes of water chemistry (Haggerty et al., 2023; Kim et al., 2021a; Shaughnessy et al., 2021; Vesselinov et al., 2018). Specifically, the results of NMF can be interpreted as endmembers and their mixing ratios (Haggerty et al., 2023; Shaughnessy et al., 2021; Vesselinov et al., 2018), which we used as a pre-processing step before applying K-means clustering. Since groundwater chemistry can be influenced by mixing and/or resulting from a series of processes, this approach reduced data dimensionality and enhanced the robustness of the subsequent clustering.

NMF decomposes the original matrix (V) into two non-negative matrices: one representing endmember compositions (H) and the other representing contributions (i.e., mixing proportion; W) of these endmembers in the context of hydrogeochemistry:

$$V = H \times W$$

The optimal number of endmembers was determined using the Elbow method, which is commonly used in clustering analysis to find the optimal number of clusters by plotting the sum of squared error (SSE) as a function of clusters (Syakur et al., 2018).

In our study, the optimal number of endmembers was identified at the minimum reconstruction error. The mixing ratios (matrix W) were subsequently used for K-means clustering. The Silhouette score—a measure of how well each data point fits within its own cluster while remaining well-separated from others—and within-cluster sum of square (WCSS) —a measure of the variability within each cluster—were used to determine the optimal cluster number. K-means clustering was repeated 50 times to achieve the highest silhouette score and the lowest WCSS.

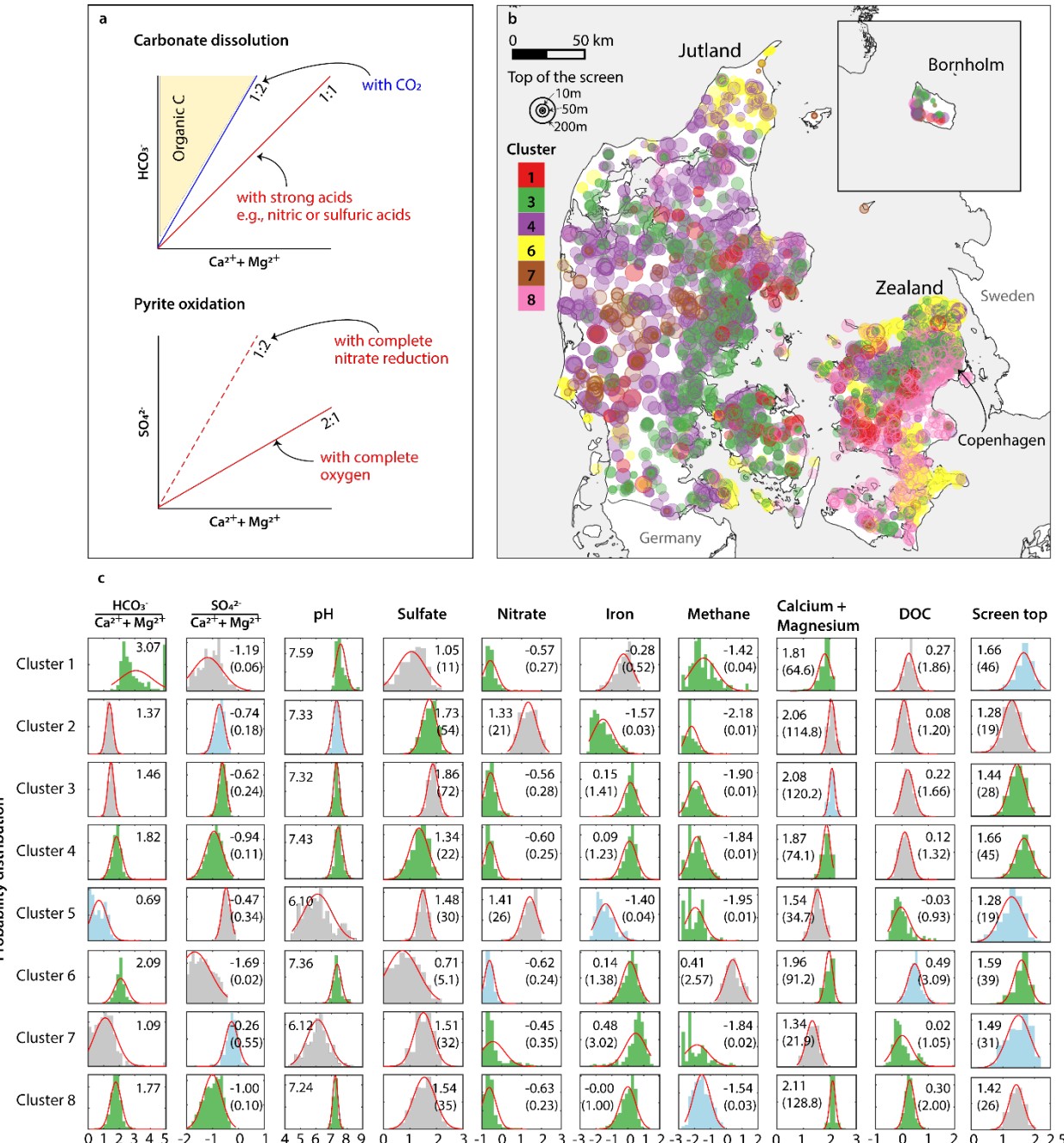

**Figure 1: Overview of the groundwater chemistry analysis results. a.** Theoretical relationships of $Ca^{2+} + Mg^{2+}$ vs. $HCO_3^-$ and vs. $SO_4^{2-}$ for key processes of the study. **b.** Map of cluster distribution. Marker colors represent different clusters, and marker sizes indicate the depth to the top of the screen. The oxic clusters (cluster 2 and 5) are not shown (available in Supplementary Figure s2). **c.** Histograms of the Non-negative Matrix Factorization (NMF) and K-means clustering results for groundwater. The red lines represent the probability distribution function of a normal distribution. The green, blue, and gray bars correspond to p-values <0.01, 0.01-0.05, and >0.05, respectively. The values displayed in the histograms are means (μ), and the numbers in parentheses are back-transformed values.

To further supervise the NMF and clustering analysis, we computed the mean stoichiometric ratios of $\frac{HCO_3^-}{(Ca^{2+}+Mg^{2+})}$ and

$\frac{SO_4^{2-}}{(Ca^{2+}+Mg^{2+})}$ at the screen level. To minimize the impact of outliers, the stoichiometric ratios were calculated by randomly
selecting 70% of the data for each screen and calculating the mean ratios 20 times. Outliers from these subsets were excluded

from the final mean ratio calculations for each screen. These two ratios can provide insights into the dominant processes and

the source of DIC. When carbonate dissolution occurs due to carbonic acid from $CO_2$, the $\frac{HCO_3^-}{Ca^{2+}+Mg^{2+}}$ ratio equals 2, with no

effect on $SO_4^{2-}$ concentrations (Figure 1a; Equation 4 Table 1). If carbonate dissolution is coupled with pyrite oxidation, the

$\frac{HCO_3^-}{Ca^{2+}+Mg^{2+}}$ ratio becomes 1, and its $\frac{SO_4^{2-}}{Ca^{2+}+Mg^{2+}}$ ratio will be either 2 (with nitrate, Equation 2 + 5 in Table 1; Figure 1a) or 0.5
(with oxygen, Equation 3+5 in Table 1; Figure 1a). When organic carbon mineralization occurs without carbonate minerals,

$HCO_3^-$ concentrations increase without changing $Ca^{2+}$ and $Mg^{2+}$ concentrations, resulting in a $\frac{HCO_3^-}{Ca^{2+}+Mg^{2+}}$ ratio greater than 2

(Figure 1a). Additionally, means of all the available constituents were also calculated at the screen level, and redox sensitive

elements such as nitrate, sulfate, iron and methane were included in the analysis to interpret the redox conditions of the clusters

(All the elements listed in Supplementary Table s1). All values were log-transformed and normalized before analysis.


## 2.3. Prediction of a national map of redox clusters at the redox interface

After identifying the redox clusters and dominant processes in groundwater, this point-scale information was upscaled to the

national scale to predict dominant electron donors of denitrification at the redox interface, defined by the interface between

the nitrate-reducing and iron-reducing zones. In Denmark, due to glaciotectonic deformation during the most recent glaciations,
the complexity of the redox architecture varies significantly, resulting in multiple redox interfaces (Kim et al., 2019; Koch et

al., 2024). Koch et al. (2024) predicted the depth to the first redox interface as well as its structural complexity at the national

scale at 25m x 25m resolution based on sediment color data and 20 explanatory variables (Table 2) using a gradient boosting

with decision tree (GBDT) algorithm (Koch et al., 2024). Using the same method and variables, we predicted a national-scale

map of redox clusters at a 100 m x 100 m resolution. GBDT is a commonly used machine learning technique in various fields
for solving prediction tasks in both classification and regression. Through iterative training, GBDT builds ensemble-based

prediction models using weak learners (i.e., decision trees). The ensemble is iteratively improved by adding decision trees that

focus on correcting the residuals of the previous model. We used Microsoft's LightGBM (Light Gradient Boosting Machine)

implementation of the GBDT algorithm in this study (Ke et al., 2017).

**Table 2. Explanatory variables for prediction of redox cluster map**

| Covariate | Description | Source |
|---|---|---|
| Clay a | Clay content (%), 0-30 cm | Adhikari et al. (2013) |
| Clay b | Clay content (%), 30-60 cm | |
| Clay c | Clay content (%), 60-100 cm | |
| Clay d | Clay content (%), 100-200 cm | |
| DEM | Elevation above sea level (m) | Danish Agency for Climate data (KDS) |
| DEM Var. | Deviation between high-resolution and low-resolution elevation (m) | |
| Slope | Surface slope gradient (deg) | |
| Geo-region | Geological regions of Denmark | Adhikari et al. (2014) |
| Landscape E. | Landscape elements of Denmark | Aarhus University - Danish Centre for Environment and Energy |
| Wetland | Wetland classification (mineral/organic) | |
| Geo. Complex. | Geological complexity | Sandersen (2021) |
| Clay thick. | Thickness of clay deposits – from surface (m) | The National Hydrological Model (DK-model, Stisen et al., 2019) |
| Sand thick. | Thickness of sand deposits – from surface (m) | |
| Aquifer thick. | Thickness of uppermost aquifer (m) | |
| Q. thick | Thickness of Quaternary deposits – from surface (m) | |
| d. to stream | Horizontal distance to stream network (m) | |
| Mean gwt. | Mean water table depth (m.b.g.l.) | |
| Max gwt. | Maximum water table depth (m.b.g.l.) | |
| recharge | Recharge (mm/year) | |

For the prediction of the redox cluster map, we first excluded groundwater screens from 1) oxic clusters (about 14% of the total screens); and 2) reduced clusters with a Silhouette score less than 0 (about 13 % of the total screens). Among the remaining screens, we selected those located near the redox interface. Considering the uncertainty of the redox interface map (Koch et al., 2024) and the availability of the groundwater chemistry data used for clustering, three subsets of groundwater chemistry data were selected: those where the depths of screen top was no more than 5 (D5), 10 (D10), and 15 (D15) meters below the redox interface. This resulted in 235 (D5), 566 (D10) and 1019 (D15) screens for training, respectively (Supplementary Figure s1). The three depth derived models were evaluated based on a 2-fold cross validation procedure, in which training two models used 50% of the data for training and validating against the remaining 50%. The model's uncertainty was quantified through

bootstrapping, by repeatedly training the classification model and predicting the redox clusters map. We generated 100 realizations of the redox cluster map for each depth criterion, each based on bootstrapped samples (with replacement).

The Shapley additive explanations (SHAP) approach was employed to assess the sensitivity of the trained classification models, i.e., feature importance (Lundberg and Lee, 2017). SHAP, based on game theory principles, explains the output of machine learning models by attributing predictions to individual covariates, quantifying their marginal contributions. In this
study, absolute SHAP values were used to measure feature importance. SHAP values are reported for each class, i.e. each cluster in the classification model, and as an average across the classes.

## 2.4. Estimation of DIC produced by denitrification in groundwater

DIC production due to denitrification was estimated by combining the predicted redox cluster maps with the average of
groundwater nitrate reduction estimates from 1990-2010 provided by the National Nitrogen Model (NKM). NKM is a comprehensive model that links three existing models (Henriksen et al., 2003; Højberg et al., 2015, 2017): 1) empirical, statistical models for reactive N leaching from the root zone (NLES; Børgesen et al., 2020); 2) the National Hydrological Model (DK-model; Stisen et al., 2019) with particle tracking in MIKE-SHE for groundwater and drain flow and nitrate reduction; and 3) statistical models for nitrate reduction in surface waters such as streams and lakes. Nitrate reduction is
simulated at the catchment level, where the average catchment size is roughly 15 km$^2$. Nitrate reduction in groundwater is calculated based on the fraction of particles passing through the redox interface, assuming instantaneous and complete denitrification at the interface. The NKM was developed, calibrated and validated using 21 years of measurements from 340 stream stations, covering approximately half of Denmark.

The groundwater nitrate reduction estimates (kg N ha$^{-1}$ yr$^{-1}$) at the catchment level, provided in shapefile format, were
converted into a GeoTIFF file using QGIS to match the extent and grid size (100m x 100m) of the redox cluster map. For each of the 100 realizations of the redox cluster maps, the groundwater nitrate reduction and cluster data were linked at the grid level using MATLAB. DIC production for each grid cell (i) was calculated by multiplying groundwater nitrate reduction (GNRi) and the stoichiometric ratio of DIC production per nitrate reduction for the respective cluster ($r_i$):

$$DIC\ produced\ by\ denitrification\ (kt\ CO_2\ y^{-1}) = \left[ \sum_{i=1}^{n} (GNR_i \times r_i \times {}^{44}/_{14} \times 10^3) \right] \times 10^{-9}$$

where $r_i$ values are 1.25 for denitrification by organic carbon and 0.33 for denitrification by pyrite oxidation coupled with carbonate dissolution (Table 1). The estimated DIC production was first summed up for the catchment level and then to the national level. The mean (μ) and standard deviation (σ) of DIC production from the 100 realizations for D5, D10 and D15, respectively, were calculated using MATLAB.

## 3. Results and Discussion

### 3.1. Redox architecture of the Danish groundwater

Our results showed that the Danish groundwater can be categorized into eight clusters): two oxic clusters (cluster 2 and 5) and six reduced clusters (1, 3, 4, 6, 7, and 8), each at various redox stages (Figure 1c). Cluster 4 was the most frequent cluster (1940 screen), followed by cluster 3 (1235 screens), cluster 8 (1012 screens), cluster 6 (747), cluster 2 (619 screens), cluster 7 (252 screens), cluster 5 (221 screens), and cluster 1 (246; Table 3). The mean silhouette score for the clustering results in this study was 0.45. Figure 1c presents histograms of the concentrations and stoichiometric ratios of input parameters. The two oxic clusters were characterized by high nitrate concentrations (mean ($\mu$) = 21 and 26 mg/L, respectively), low iron concentrations ($\mu$ = 0.03 and 0.04 mg/L, respectively) and relatively shallow depths ($\mu$ = 19 m for both; Figure 1c). These clusters had $\frac{HCO_3^-}{Ca^{2+}+Mg^{2+}}$ ratios of 1.37 (cluster 2) and 0.67 (cluster 5), suggesting a strong influence of anthropogenic impact, such as carbonate dissolution by nitric acid produced by microbial oxidation of ammoniacal fertilizers as mentioned earlier (Barak et al., 1997; Perrin et al., 2008). Cluster 2 was found mainly in the eastern part of Denmark while cluster 5 were mainly in the western and northern regions (Supplementary Figure s2). Denitrification in these oxic clusters was considered negligible; therefore, further interpretations focused on the six reduced clusters.

The six reduced clusters have nitrate concentrations below 1 mg/L and elevated iron concentrations ($\mu$ = 0.52-3.02 mg/L; Figure 1c). Cluster 1 and 6 had high $\frac{HCO_3^-}{Ca^{2+}+Mg^{2+}}$ ratios ($\mu$ = 3.07 and 2.09, respectively; Figure 1c), indicating the dominance of organic carbon oxidation. Sulfate concentrations varied widely in both clusters, but the mean values were the lowest among all the reduced clusters ($\mu$ =11 and 5 mg/L, respectively; Figure 1c), indicative of sulfate reduction. Notably, cluster 6 had high methane concentrations ($\mu$ = 2.57 mg/L; Figure 1c), suggesting highly reduced methanogenic conditions.

Cluster 3 had the highest sulfate concentrations ($\mu$ = 72 mg/L) and the second-highest Fe concentrations ($\mu$ = 1.41 mg/L; Figure 1c). The elevated sulfate levels were attributed to pyrite oxidation, and its elevated Fe concentration could be due to incomplete pyrite oxidation and/or reductive dissolution of Fe(II)-bearing minerals. While anthropogenic sulfur input can elevate sulfate concentrations in groundwater, cluster 3 displayed higher sulfate concentrations than the oxic clusters ($\mu$ = 54 and 30 mg/L, respectively). Since the oxic clusters were generally closer to the direct input of groundwater recharge and thus more responsive to anthropogenic signals, we conclude that pyrite oxidation likely plays a larger role in the elevated sulfate concentrations in this cluster. However, the $\frac{SO_4^{2-}}{Ca^{2+}+Mg^{2+}}$ ratios were lower than the theoretical values for pyrite oxidations ($\mu$ = 0.24). This discrepancy may arise if carbonate mineral dissolution occurs by both strong (nitric and sulfuric acids) and carbonic acids (H$_2$CO$_3$; $\frac{HCO_3^-}{Ca^{2+}+Mg^{2+}}$ = 2) as indicated by the $\mu$ of the $\frac{HCO_3^-}{Ca^{2+}+Mg^{2+}}$ ratio of 1.48. This value aligns with ratios typical of agricultural streams (Perrin et al., 2008; Stets et al., 2014), supporting our interpretation.

Cluster 4 was characterized by low sulfate concentrations ($\mu$ = 22 mg/L) and found at greater depths ($\mu$ = 45 m). These conditions indicated moderately reduced states, likely transitioning from Fe-reducing to sulfate reducing conditions. The high

$\frac{HCO_3^-}{Ca^{2+}+Mg^{2+}}$ ratios ($\mu$ = 1.82) and relatively low $Ca^{2+} + Mg^{2+}$ concentrations ($\mu$ = 74 mg/L) were attributed to $HCO_3^-$ production from the oxidation of organic carbon by reducing sulfate and Fe-oxides in groundwater already in equilibrium with carbonate minerals (e.g. calcite).

Cluster 7 showed the highest Fe concentrations ($\mu$ = 3.02 mg/L) and lowest pH ($\mu$ = 6.12) among the reduced clusters (Figure 1c). Its mean $\frac{SO_4^{2-}}{Ca^{2+}+Mg^{2+}}$ and $\frac{HCO_3^-}{Ca^{2+}+Mg^{2+}}$ ratios were 0.55 and 1.09, respectively, closely aligning with those of pyrite oxidation with oxygen coupled with carbonate dissolution (Equation 3 and 5 in Table 1). Previous transect-level studies from the area near cluster 7 documented that denitrification in groundwater is mediated by pyrite oxidation (Jessen et al., 2017; Postma et al., 1991). They also reported that the study area is depleted with carbonates and attributed total inorganic carbon (TIC) in groundwater to agricultural liming. The low pH and $Ca^{2+} + Mg^{2+}$ concentrations of cluster 7 are consistent of these studies' results. However, cluster 7 displayed low sulfate ($\mu$ = 32 mg/L) but high Fe ($\mu$ = 3.02 mg/L) concentrations for conditions dominated by complete denitrification by pyrite oxidation (Equation 2 in Table 1). For instance, Danish oxic groundwater displays between 40-50 mg/L of nitrate (Thorling et al., 2024). If it is denitrified by pyrite oxidation, it will result in an increase of $SO_4^{2-}$ concentrations by 41-51 mg/L and very low Fe concentrations. We attributed these discrepancies to either/or combinations of 1) sulfate and iron reduction by oxidation of organic carbon; 2) mixing with reduced groundwater from deeper depths; and/or 3) incomplete oxidation of $S^{-1}$ of pyrite (Zhang et al., 2009):

$$20FeS_2 + 14NO_3^- + 44H^+ \rightarrow 20\ Fe^{2+} + 5SO_4^{2-} + 35S^0 + 7N_2 + 22H_2O$$

The low $\frac{HCO_3^-}{Ca^{2+}+Mg^{2+}}$ ratios of cluster 7 imply that the role of organic carbon decomposition may be minor; thus, the second and/or third processes may more likely be responsible for the cluster 7 groundwater chemistry.

Cluster 8 exhibited signs of transitioning from Fe- to sulfate-reducing/methanogenic environments in organic- and carbonate-rich conditions. The $\frac{HCO_3^-}{Ca^{2+}+Mg^{2+}}$ ratio ($\mu$ = 1.7) was significantly higher than for carbonate dissolution by strong acids. Moderately elevated dissolved organic carbon (DOC; $\mu$= 2.0 mg/L) and detectable methane ($\mu$= 0.03 mg/L) indicated an abundance of organic C. Additionally, this cluster exhibited the highest $Ca^{2+} + Mg^{2+}$ concentrations among all clusters, further supporting carbonate-rich conditions.

**Table 3. Summary of cluster analysis and redox cluster prediction**

| Cluster | Number of screens | Predicted area at the redox interface (km²) | Redox stage | Dominant electron donor for denitrification |
|---------|-------------------|---------------------------------------------|-------------|---------------------------------------------|
| 1 | 246 | 195 | Sulfate-reducing | Organic C |
| 2 | 619 | - | Oxic | No denitrification |
| 3 | 1235 | 26,457 | Fe-reducing | Pyrite |
| 4 | 1940 | 6,027 | Close to sulfate-reducing | Organic C |

| 5 | 221 | - | Oxic | No denitrification |
|---|---|---|---|---|
| 6 | 747 | 743 | Methanogenic | Organic C |
| 7 | 252 | 6,342 | Fe-reducing | Pyrite |
| 8 | 1012 | 3,209 | Fe- and sulfate-reducing to methanogenic | Organic C |

Altogether, the redox sequence of the clusters can be summarized according to the redox ladder: cluster 2 and 5 as oxic, cluster 3 and 7 as Fe-reducing, cluster 4 and 8 as transitioning from Fe-reducing to sulfate reducing, cluster 1 as sulfate-reducing, and cluster 6 as methanogenic (Table 3). We hypothesized that if cluster 3 and 7 are found near the redox interface, denitrification in these regions would likely be mediated by pyrite oxidation. Both clusters displayed pyrite oxidation signals and completion of nitrate reduction, which are expected at the redox interface. Conversely, if cluster 1 and 6, representing highly reduced conditions, are found at the redox interface, denitrification in these areas may be mediated by organic carbon oxidation. These clusters are expected to appear in areas with abundant organic matter, where the thicknesses of nitrate-reducing and iron-reducing zones may be too thin to resolve at the scale of most of the groundwater screens. Cluster 4 and 8, interpreted as transitioning to Fe- and sulfate reducing/methanogenic conditions, did not display clear dominance of neither pyrite oxidation nor organic C oxidation. However, we suggest that organic C oxidation may be more dominant in these clusters as indicated by their $\frac{HCO_3^-}{Ca^{2+}+Mg^{2+}}$ ratios compared to those clusters where pyrite oxidation was the most probable process.

### 3.2. A national map of redox clusters at the redox interface

Figure 2a shows the predicted spatial distribution of redox clusters at the redox interface using the 10 m (D10) distance criterion. Results for D5 (5 m) and D15 (15m) are available in Supplementary Figures s1. The overall accuracies based on the 2-fold cross validation were 0.58 (D5), 0.61 (D10), and 0.64 (D15), respectively. The final maps (Figure 2a and Supplementary Figure s1) of redox clusters were generated by GBDT models trained on 100% of the available data.

The redox cluster map resembles the landscape elements of Denmark (Figure 2c), although the contribution of landscape classification to the prediction was minimal except for cluster 7 (Figure 2b). Among the 20 explanatory variables, the thickness of Quaternary deposits and groundwater recharge were identified as the two most influential variables (Figure 2b). These findings highlight the central role of hydrogeology in determining the distribution of clusters at the redox interface. For instance, cluster 7, representing nitrate-reduction by pyrite oxidation in carbonate-limited environments, was predicted predominantly in meltwater plain areas south and west of the main stationary line of the ice sheet from the last glaciation. In contrast, cluster 3, interpreted similarly but under carbonate-rich conditions, was mainly found east and north of the main stationary line. This difference likely arises because the meltwater plains west and south of the main stationary line have been exposed to weathering for longer than areas covered by the last ice sheets, leading to the depletion of easily weatherable

minerals such as carbonates. Additionally, groundwater recharge emerged as the most influential predictive variable for cluster 7, further suggesting intensive weathering. Overall, cluster 7 corresponds to more intensively weathered hydrogeological conditions.

In the postglacial (more recent) sediment areas, where fresh organic matter is likely abundant, highly reduced conditions were predicted around the redox interface, which is consistent with our hypothesis of the cluster distribution at the redox interface. Cluster 6, associated with methanogenesis, would require reactive organic matter, which is consistent with its presence in dunes and postglacial marine sediment areas (Hansen et al., 2001; Jakobsen and Cold, 2007; Jakobsen and Postma, 1999). The thickness of the Quaternary deposits and the mean depth of the groundwater table were the most important predictors for cluster 6, implying that the combination of organic-rich and a shallow groundwater table can result in such extremely reduced conditions. Cluster 1 (sulfate reducing conditions) rarely appeared near the redox interface.

The spatial distributions of cluster 4 and cluster 8, which transition from Fe- to sulfate-reducing conditions, were also distinctive. Cluster 4 was mostly found in postglacial marine sediments and meltwater plains within the main stationary line. These meltwater plains typically evolve into river valleys (Kaiser et al., 2007), where recent reactive organic matter accumulates. While cluster 8 was predicted in areas where a thin Quaternary layer overlays limestone in Zealand. This limestone is primarily the Danian bryozoan limestone, formed in cool and deep-water conditions shortly after the Cretaceous-Tertiary (K/T) mass extinction (Bjerager and Surlyk, 2007; Jorgensen, 1988). The limestone represents a highly diverse marine benthic ecosystem, rich in organic matter such as bryozoan-coral mounds (Bjerager and Surlyk, 2007; Jorgensen, 1988), leading to formation of pyrite in the limestone from sea water sulfate. Indeed, in bryozoan limestone, pyrite is observed (Damholt et al., 2006). In both cluster 4 and 8, pyrite oxidation may be probable. However, the groundwater chemical signatures (i.e., $\frac{HCO_3^-}{Ca^{2+}+Mg^{2+}}$ ratios and highly reduced conditions) of these two clusters may indicate a more dominant role of organic C oxidation.

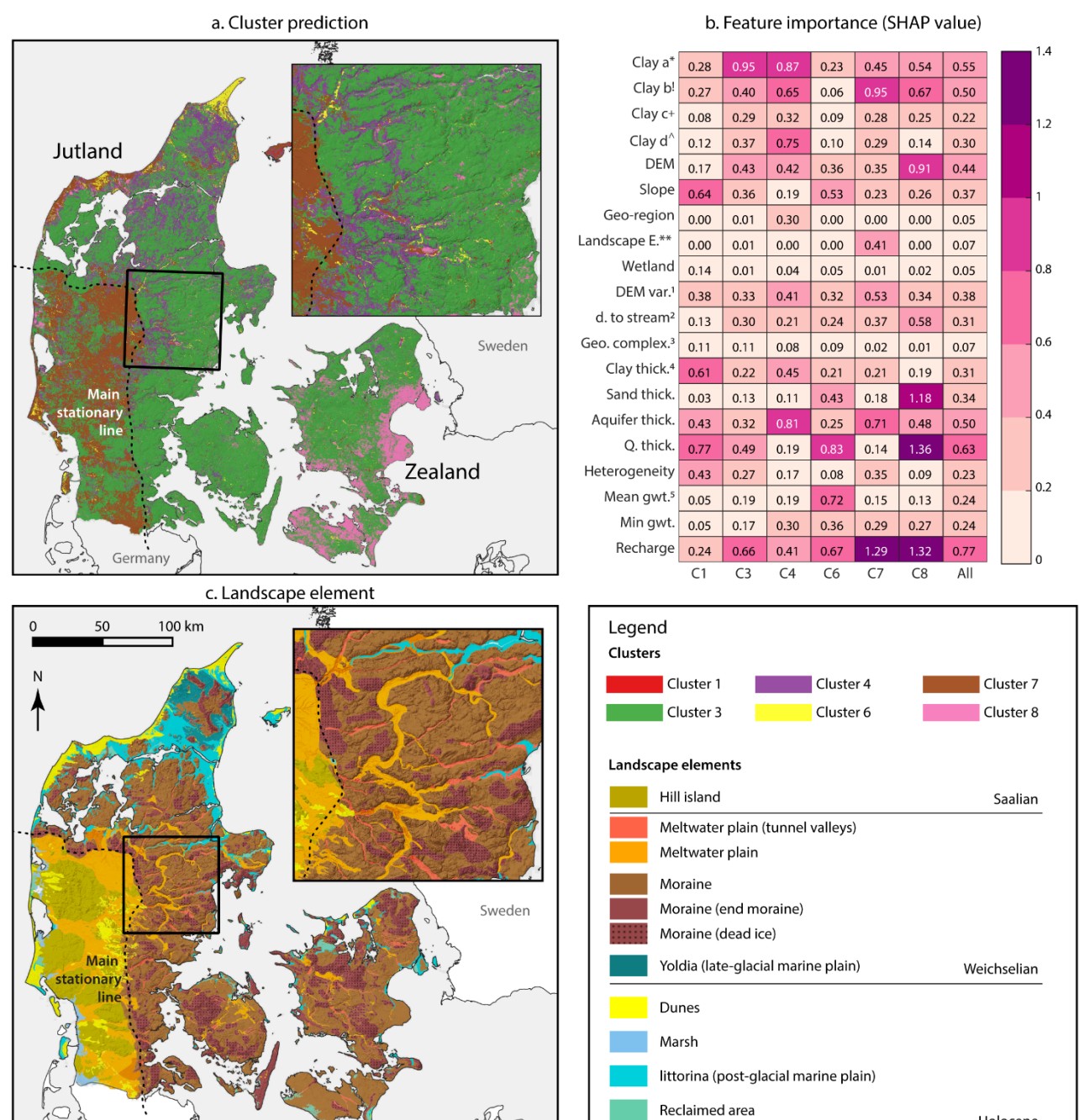

**Figure 2. a.** Map of redox clusters at the redox interface with the distance criterion 10 m from the redox interface to the screen tops (D10). **b.** Feature importance of the 20 input variables used for map prediction. *Clay a: clay content (%) 0-30 cm; !Clay b: Clay content (%) 30-60 cm; +Clay c: clay content (%) 60-100 cm; ^Clay d: clay content (%), 100-200 cm;**Landscape E.: Landscape Element; [1]DEM var.: Digital Elevation Map variability; [2]d. to stream: horizontal distance to stream network (m); [3]Geo. Complex.: geological complexity; [4]Clay thick.: thickness of clay deposits from the surface (m). Sand thick., Aquifer thick., and Q. thick. refer to the thickness of sandy layer, aquifer, and quaternary layers, respectively; [5]Mean gwt.: mean depth to groundwater table in meters below the ground level. **c.** Landscape element map of Denmark with the legend displayed on the right.

320

325

We compared our predicted redox clusters with findings from previous Danish studies on redox processes at both transect and catchment scales (Jakobsen and Cold, 2007; Jakobsen and Postma, 1999; Kim et al., 2021a, b; Postma et al., 1991; Figure s3). For example, Postma et al. (1991) investigated denitrification along a transect following a groundwater flow path in an unconfined sandy aquifer in Denmark. They found that nitrate concentration decreased rapidly at the redoxcline, primarily by pyrite oxidation, despite the higher abundance of organic matter. This transect corresponds to cluster 7 in our classification, consistent with our prediction.

Jakobsen and Postma (1999) also conducted a transect-based field study along a groundwater flow path within the dune area underlain by postglacial sand of central Rømø, Denmark. They investigated how redox processes including iron reduction, sulfate reduction, and methanogenesis vary horizontally and vertically. They concluded that although slow, the fermentation of organic matter controls the co-occurrence of multiple redox processes. Jakobsen and Cold (2007) reported similar findings to Jakobsen and Postma (1999) in an aeolian/post-glacial marine sandy aquifer in northern Zealand. Our redox cluster map indicated that Rømø is primarily predicted as cluster 6 (methanogenesis) and cluster 7 (Fe-reducing), while northern Zealand near the Jakobsen and Cold study site is classified as cluster 4 (close to sulfate reducing), demonstrating reasonable agreement with the earlier transect-based findings. The key role of organic matter fermentation in these study areas further suggests the dominance of organic carbon oxidation as the electron donor for denitrification.

At the catchment scale, Kim et al., (2021a) investigated the subsurface structure of denitrification zone in a glacial sediment catchment in Northern Jutland, using a combination of geophysical, geological, hydrological, and geochemical data. By analyzing groundwater chemistry data using K-means clustering analysis, they found that both pyrite oxidation and organic carbon oxidation contribute to denitrification in shallow groundwater. Note that the data used in Kim et al (2021a) were not included in our analysis. They showed that the chemistry of reduced groundwater near the stream showed clear signatures of organic carbon oxidation, while that in the rest of the catchment indicated pyrite oxidation. Consistently, our results identified cluster 4 along the stream, while the remainder of the catchment was predominantly classified as cluster 3 (Fe-reducing, pyrite oxidation; Table 3). Kim et al., (2021b) also carried out a similar study in eastern Jutland in a clay-till catchment. High resolution profiles of groundwater geochemistry revealed that denitrification in this catchment may be primarily driven by pyrite oxidation. This catchment was also predominantly predicted as cluster 3. Overall, our predictions of redox clusters and dominant electron donors for denitrification showed strong agreement with the results of prior process-focused field investigations.

### 3.3. Anthropogenic DIC production in groundwater due to N fertilizer applications

For the period 1990-2010, the NKM estimated that an average of 125 kt of N $yr^{-1}$ was reduced in groundwater on the national scale. At the catchment scale, higher groundwater nitrate reduction was observed in western Jutland, particularly in the

southeast, with values up to 86 kg N ha$^{-1}$ yr$^{-1}$ (Figure 3a). In contrast, in Zealand, the nitrate reduction in groundwater ranged from 0 to 10 kg N ha$^{-1}$ yr$^{-1}$ (Figure 3a). These regional variabilities can be attributed to hydrogeological differences. In eastern Jutland and Zealand, drain flow is the main reactive N pathway (Møller et al., 2018), whereas in western Jutland, the sandy soils result in lower drain flow, and reactive N is instead transported via groundwater (Møller et al., 2018). In addition, reactive N leaching in western Jutland is relatively high due to intensive animal farming and high infiltration in the sandy soils. As a result, groundwater's contribution to nitrate reduction at the catchment level becomes more pronounced in western Jutland.

We estimated that DIC production in groundwater by denitrification of nitrate leached from the agricultural soils, for the D5, D10, and D15 models, was 182 (standard deviation (SD): 10.58), 204 (SD: 11.25), and 229 (SD:10.50) kt of $CO_2$ yr$^{-1}$, respectively. Like nitrate reduction, $CO_2$ production was characterized by spatial heterogeneity. Although western Jutland showed the highest nitrate reduction in groundwater, DIC production was moderately high, ranging from 60 to 145 kg $CO_2$ ha$^{-1}$ yr$^{-1}$. While the highest DIC production was predicted in northern Jutland (up to 180 kg $CO_2$ ha$^{-1}$ yr$^{-1}$; Figure 3b) despite the low to moderate nitrate reduction in groundwater (< 60 kt of N yr$^{-1}$). Such results can be attributed to differences in dominant electron donors. In the northern Jutland, denitrification was predicted to be mediated by organic C (Figure 3a). Compared to pyrite oxidation, which releases 0.33 moles of DIC per mole of nitrate reduction, organic matter-mediated denitrification increases 1.25 moles of DIC per mole of nitrate reduction (Table 1). Consequently, more DIC, thus more $CO_2$, is produced by denitrification in this region.

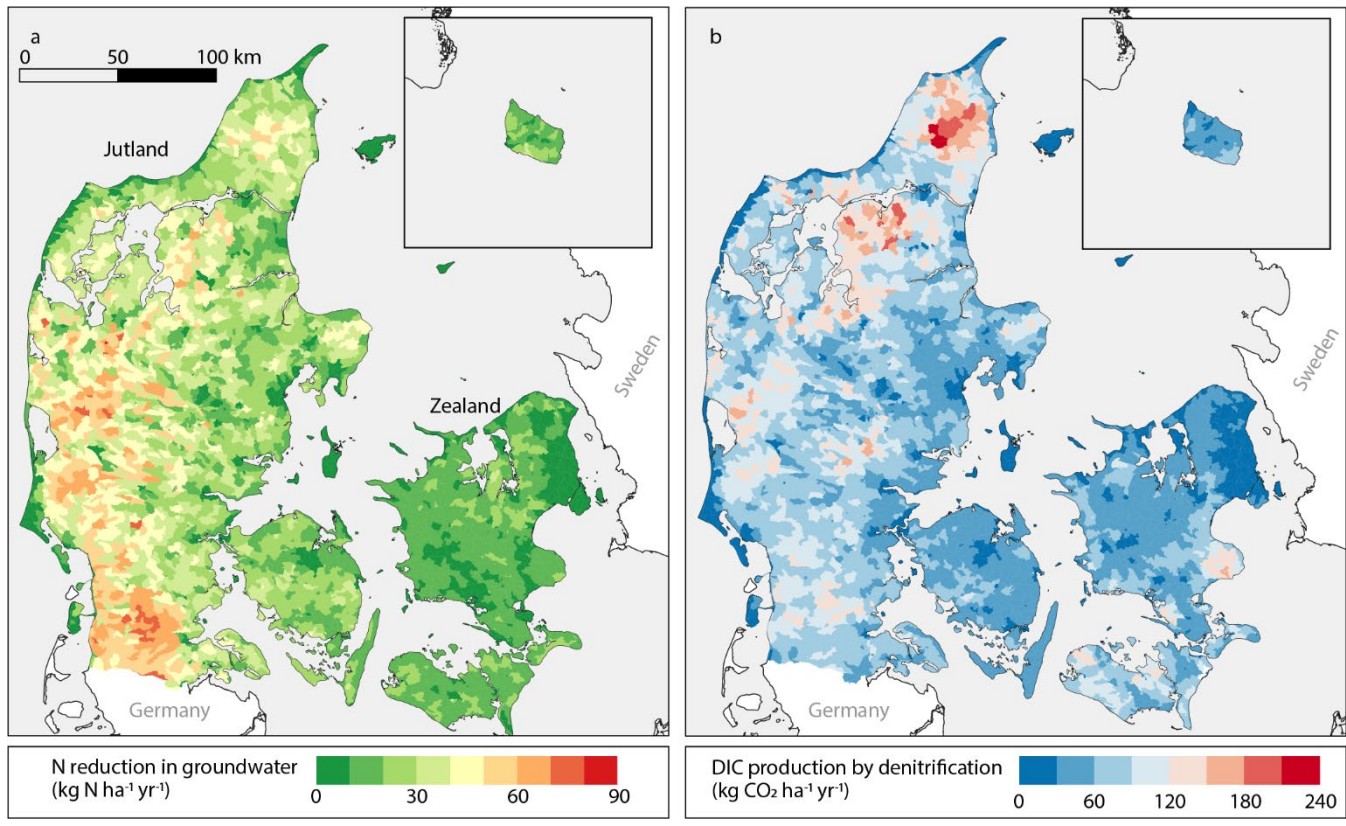

**Figure 3. Maps of estimates of (a) nitrate reduction in groundwater in kg N ha$^{-1}$ yr$^{-1}$ and (b) DIC production by denitrification at the catchment scale in kg CO₂ ha$^{-1}$ yr$^{-1}$.**

### 3.4. Contribution of denitrification in the national inventory of greenhouse gas emissions

Agriculture in Denmark is the second-largest contributor to greenhouse gas emissions excluding land use, land-use change and forestry (LULUCF; Nielsen et al., 2022). According to the IPCC guidelines, the current national inventory for the agricultural sector includes emissions from 1) $CH_4$ due to enteric fermentation, manure management, and field burning; 2) $N_2O$ from manure management, N-fertilizer use in agricultural soils and field burning; and 3) $CO_2$ from liming, urea, and other C-containing fertilizers (Nielsen et al., 2022). In 2020, the agricultural sector of Denmark emitted a total of 11,268 kt CO₂-eq. yr$^{-1}$ GHGs, primarily as $CH_4$ (5,881 kt CO₂-eq. yr$^{-1}$) and $N_2O$ (5,132 kt CO₂-eq. yr$^{-1}$; Figure 4). Carbon dioxide accounted for about 2 % (254 kt CO₂-eq. yr$^{-1}$) of the total GHG emissions from agriculture (Figure 4). The largest source of $CO_2$ was liming (250 kt of CO₂-eq. yr$^{-1}$), followed by other C-containing fertilizers (4 kt of CO₂-eq. yr$^{-1}$) and urea (1 kt of CO₂-eq. yr$^{-1}$).

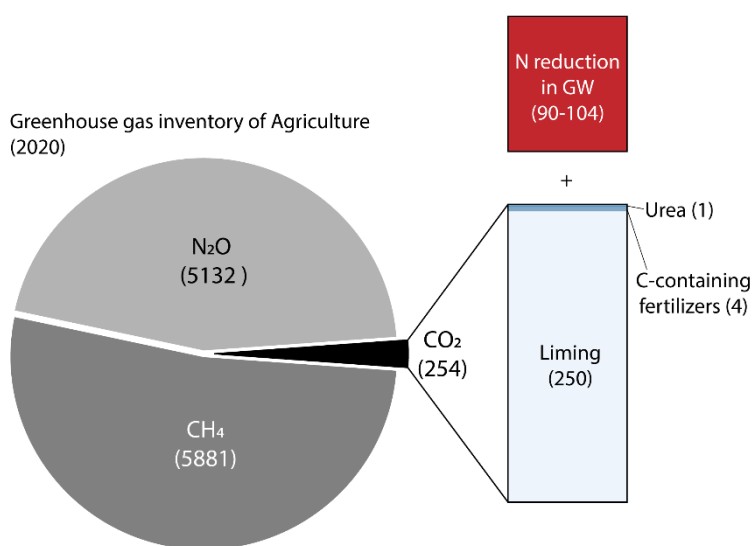

**Figure 4. National inventory of greenhouse gas emissions from Danish agriculture. The numbers represent GHG emissions in kt of $CO_2$ equivalent $yr^{-1}$. The red box labeled "N reduction in GW" shows the potential $CO_2$ emissions from denitrification in groundwater estimated in this study.**

Our study estimated that approximately 204 kt of $CO_2$-eq. $yr^{-1}$ of DIC is produced by denitrification in groundwater. As groundwater discharges back into surface waters such as streams, $CO_2$ will be degassed from groundwater because groundwater is supersaturated with $CO_2$ with respect to the atmosphere. The actual magnitude of $CO_2$ emissions from groundwater depends on various physicochemical conditions, particularly the degree of calcite saturation of groundwater as $CO_2$ degasses. Among the groundwater screens used for the D10 prediction, 476 of them had field measured pH values. Calculated using PHREEQC (Parkhurst and Appelo, 2013), their $\log(pCO_2)$ levels ranged between -2.14 (7,244 µatm) to -1.49 (32,359 µatm; Figure 5a), and saturation indices (SI) for calcite ($SI_{CaCO3}$) were close to zero except for cluster 7 (-3.72; Figure 5b). Assuming equilibration with atmospheric $pCO_2$ (400 µatm), $SI_{CaCO3}$ of these groundwater screens would increase by approximately 1-2 order of magnitude (Figure 5c). These results suggest (based on Table 1) that for cluster 1, 3, 4, 6, and 8, approximately half of the DIC produced by denitrification will be emitted into the atmosphere with the remainder being restored as calcite. We further quantified the distribution of the increased DIC between $CO_2$ emissions, calcite precipitation and the solution using PHREEQC by taking cluster 2 (oxic cluster) as initial conditions (Supplementary Text s1). PHREEQC results indicated that half of the increased DIC was emitted as $CO_2$, while the remainder was precipitated as calcite (30 to 66% of the increased DIC) or stayed in solution as bicarbonate (-15 to 10% of the increased DIC).

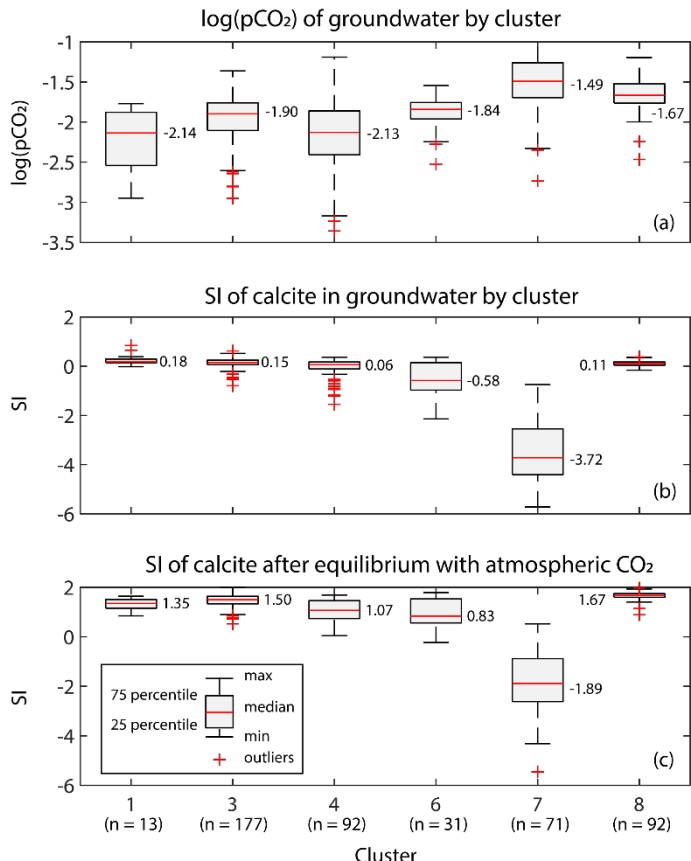

**Figure 5. Box plots of (a) log of pCO₂ of groundwater by cluster, (b) saturation index (SI) of calcite in groundwater by cluster, (c) SI of calcite in groundwater after equilibrium with atmospheric CO₂ by cluster.**


In cluster 7, however, carbonate may have originated primarily from liming, meaning its $CO_2$ emissions were already accounted as liming; thus, no $CO_2$ will be emitted due to denitrification. On the other hand, cluster 7 groundwater could pass through the carbonate front, becoming saturated with calcite before discharging into the stream. In this case, its ratio of $CO_2$ emission per nitrate reduction will be the same as cluster 3. Taking these two cases as the minimum and maximum limits, we

estimated that 90-104 kt of $CO_2$ yr$^{-1}$ will be emitted due to denitrification in groundwater. It is important to note that $CO_2$ production from denitrification in streams and estuaries was not included in this estimation. For comparison, Martinsen et al. (2024), using machine learning models with observational data and hydrological model outputs, estimated that about 513 kt of $CO_2$ yr$^{-1}$ is released from the national stream network (Martinsen et al., 2024), and the specific effect of nitrate in this is not given.

Our results indicate that groundwater denitrification may represent a significant anthropogenic source of $CO_2$—comparable in magnitude to liming and substantially larger than other $CO_2$ sources currently included in the IPCC guidelines. These findings imply that current estimations of $CO_2$ emissions from the agricultural sector may be underestimated. While further evaluation

is needed, our findings suggest that $CO_2$ emissions from denitrification should be considered in future revision of the IPCC GHG inventory guidelines. These results would benefit from validation through additional studies across diverse settings.

Denitrification is one of the most extensively investigated biogeochemical processes globally, and findings from these studies may help estimate $CO_2$ emissions from denitrification in groundwater, and potentially in streams and estuaries under varying agricultural, climatic and geological conditions. By synthesizing existing research, $CO_2$ emission factors for denitrification could be more accurately constrained at both local and national levels.

Nitrate pollution is a major environmental issue around the world. Research and policy efforts have primarily focused on water
quality impact, such as eutrophication and public health concerns. However, our study underscores the need to pay attention to the climatic consequences of nitrate pollution as well as N fertilizer use and management. Remediation and restoration efforts for nitrate pollution will inevitably lead to anthropogenic $CO_2$ emissions. Thus, more holistic approaches are necessary to address both water quality and climate impact.

**4. Conclusion**

In this study, we evaluated the role of groundwater denitrification of nitrate from agricultural N use as a potentially important anthropogenic net source of $CO_2$ from agriculture at the national scale. Using long-term dataset of groundwater chemistry in Denmark, we characterized the subsurface redox architecture and identified the dominant denitrification processes. This point-scale information was then scaled up to the national level to produce a predictive map of redox clusters at the redox interface,
which could also provide information on the dominant electron donor of denitrification. The redox cluster map highlighted the critical role of hydrogeology in shaping dominant processes by controlling the availability of inorganic carbon and types of reduced materials i.e., pyrite and organic carbon. By integrating these findings with the NKM estimates of nitrate reduction in groundwater, we calculated that denitrification contributed about 205kt of $CO_2$ as DIC in groundwater annually, with about half of this $CO_2$ likely released into the atmosphere when groundwater discharges into surface waters.

Our results indicated that, in Denmark $CO_2$ emissions from groundwater denitrification are comparable in magnitude to those from liming, a predominant source of $CO_2$ in the agricultural sector. To the best of our knowledge, this is the first study to quantify the contribution of groundwater denitrification to atmospheric $CO_2$ on a national scale. Besides liming, the current IPCC guidelines on greenhouse gas inventory account $CO_2$ emissions from urea and C-containing fertilizers, which are up to two orders of magnitude smaller than those from denitrification in groundwater in Denmark. These findings suggest that the
current $CO_2$ emissions from the agriculture sector are likely underestimated, and that subsurface denitrification may be a non-negligible component. While $CO_2$ is a relatively minor component of the overall agricultural GHG budget, our findings highlight that groundwater denitrification represents a previously unaccounted anthropogenic $CO_2$ source. We recommend that this process should be considered in future efforts to improve the completeness of agricultural $CO_2$ inventories.

This study also highlights the value of integrating process-based understanding with data-driven methods to address the challenges posed by spatial heterogeneity and the upscaling of complex subsurface biogeochemical processes such as greenhouse gas emissions. While the mechanisms and primary controls of denitrification have been extensively studied at the small spatial scales such as profiles, transects, and catchment, translating this knowledge into robust, large-scale quantification has remained challenging. By integrating insights from hydrogeology, groundwater redox chemistry, and long-term monitoring data within a predictive mapping framework, we demonstrated how multidisciplinary approaches—including machine learning—can integrate fundamental process-based understanding. This integrative approach offers a promising pathway not only for improving nitrate management strategies but also for reducing uncertainties in greenhouse gas inventories from agricultural systems and more generally, for large scale studies on groundwater geochemistry.

Denitrification is a natural process and is necessary for mitigating nitrate pollution in groundwater. However, it inevitably releases $CO_2$ through the mineralization of organic C or carbonate minerals. These geological C sources would otherwise remain stable over geological timescales. Restoring these losses will be challenging. Therefore, strategies addressing N pollution in aquatic ecosystems, particularly in groundwater, must consider both water quality management and climate impact comprehensively.

**Data availability**

All groundwater chemistry data is available on the National Borehole Database (www.geus.dk/jupiter). The data for predicting the national map is available under following DOI: 10.22008/FK2/I9YCF6 (Koch, 2004).

**Author contribution**

HK, JK, BGH, and RJ contributed to conceptualization. HK and JK conducted analyses. HK acquired funding and managed the project. HK wrote the original draft and revised it with comments from all coauthors.

**Acknowledgements**

This project was funded by Geocenter, Denmark. We thank two anonymous reviewers for providing constructive feedback. We thank Anker Lajer Højberg for providing the results of NKM.

485 **Financial support**

This project was funded by Geocenter, Denmark: Emission of geologic C by agricultural nitrate leaching – an overlooked $CO_2$ source in terrestrial ecosystems? (2021)

**Competing interests**

The authors declare that they have no conflict of interest.

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
