# Peer review of "A national scale redox clustering for quantifying CO2 emissions from groundwater denitrification"

_EGUsphere, 2024_

## Author Comment (AC1)

**Reviewer 2**

The manuscript "Nitrate reduction in groundwater as an overlooked source of agricultural $CO_2$ emissions" provided an estimation of CO2 emissions by heterotrophic Denitrification from groundwater, based on monitoring data of Danish groundwater and come up with the hypothesis that heterotrophic Denitrification is a significant source for DIC and outgassing CO2 and should be taken into account for the national GHG emissions estimations and that this may also important for other countries.

Generally, to conduct an entire GHG budget the emissions from groundwater have to be taken into account and a robust estimation is necessary, nevertheless some questions arise.

- At which point and time the exchange of the GHG including CO2 between groundwater and atmosphere happened. After the leaching to river systems and marine waters, or at a earlier point. So maybe to CO2 from submarine groundwater discharge (SGD) already count for the general GHG budget

This is an important question. Our study specifically focused on the subsurface processes, aiming to quantify CO2 emissions resulting from denitrification in groundwater. However, we fully acknowledge the fate of DIC in groundwater as it moves into different water bodies such as streams, lakes, and coastal areas is complex and requires further investigation. Degassing of CO2 in streams, lakes, and coastal waters is influenced by multiple factors such as water turbulence, wind speed, relative humidity and temperature. However, CO2 degassing from groundwater will likely occur when groundwater discharges into surface waters i.e., streams, wetlands, riparian zones, where it comes into direct contact with the atmosphere. It is largely driven by the fact that groundwater's pCO2 levels are significantly higher than atmospheric CO2 concentrations, creating a natural gradient that promotes CO2 degassing to reach equilibrium. In the revised manuscript, we will add a few sentences explaining where CO2 in groundwater will be emitted into atmosphere.

We acknowledge that in Denmark, we have not quantified the contribution of submarine groundwater discharge (SGD) to the national water budget. While SGD might be an important contributor at a local scale water and nutrient budgets, its contribution at larger scales will be minor. Therefore, we concluded that SGD is a minor pathway to release CO2 emissions from denitrification in groundwater.

- You just mentioned heterotrophic denitrification, because that produced CO2. But what is the percentage of autotrophic denitrification in the systems and does that play a role for CO2 fixation?

In this study, we considered both heterotrophic (i.e., organic carbon mediated) and autotrophic (i.e., pyrite-oxidation mediated) denitrification. These two processes have been identified as the dominant denitrification reactions in Danish groundwater as well as in other regions with similar geological setting.

In case of autotrophic denitrification by pyrite oxidation, the process contributes to an increase of DIC. This occurs because pyrite oxidation generates protons, which promotes calcite dissolution if calcite is present. While some other form of autotrophic denitrification processes that fix CO2 (such as driven by oxidation of sulfur or H2) occur under more strongly reduced conditions. Our analysis showed that these reactions may be limited in spatial extent at the redox interface. For instance, Cluster 1 (sulfur-reducing conditions) and 6 (methanogenesis) were predicted to account for only 0.5 % and 2% of the total area at the redox interface. We interpreted that these conditions typically occur in organic-rich environments, where organic-mediated denitrification would likely dominate and reduce nitrate at shallower depth before groundwater reaches these deeper, more reduced zones.

In the revised manuscript, we will provide a table of summarizing cluster prediction results including predicted areas, numbers of screens, redox conditions of each cluster, and dominant denitrification processes. This table will provide information of the relative importance of autotrophic and heterotrophic denitrification.

- Anaerobic denitrification also produced TA, how that was taken into account and maybe increase the capability to store DIC and emit less. See: Middelburg, J. J., Soetaert, K., and Hagens, M.: Ocean Alkalinity, Buffering and Biogeochemical Processes, Reviews of Geophysics, 58, e2019RG000681, https://doi.org/10.1029/2019RG000681, 2020.

Yes, both heterotrophic and autotrophic denitrification reactions considered in this study contribute to an increase in dissolved inorganic carbon in groundwater. The increase in DIC also elevates pCO2 in groundwater, resulting in CO2 degassing when the supersaturated groundwater encounters the atmosphere. Therefore, while denitrification does temporally increase DIC in groundwater, this acts as a short-term C storage, typically on the order of years to decades, before the CO2 is eventually released into the atmosphere. It is also important to highlight that denitrification in groundwater mineralize and mobilize both organic and inorganic C pools that would otherwise remain stored over a geological time scale.

The suggested reference, on the other hand, focuses on total alkalinity in ocean systems, where the residence times of both water and carbon are significantly longer than in groundwater systems. This extended residence times allows for long-term carbon storage and buffering in the ocean, which contrast with the more dynamic and short-term nature of the C cycle in

groundwater.  Therefore, while the oceanic context provides valuable insights into global carbon cycling and buffering, its direct comparison to groundwater systems may not be fully applicable. Therefore, we will not implement any changes to the revised manuscript.

Some specific comments:

| | |
|---|---|
| L 18/19: Why you mention CO2-eq. and not just CO2 although is it as DIC. Where and when the 50% emitted to the atmosphere | It is displayed as CO2-eq because this number can be compared to the amount of CO2 emissions. Therefore, we will keep CO2-eq as the unit to express DIC. The second question was addressed above. |
| L 35/41: this paragraph raises some questions. Nitrogen fertilizers are more than nitrate, so that also organic nitrogen and ammonium is part of that. So in consequence nitrification plays also a crucial role and can be a significant source of N2O. The references for the N2O sources Ritchie at al, 2023 just focused on "anthropogenic" Sectors. So that natural processes and sources with is maybe also anthropogenic impacted are not negligible. Especially ODZ and also groundwater discharge can be source of N2O and other GHGs | "Nitrogen fertilizers" was used to infer both synthetic fertilizers and organic fertilizers as well i.e., manure.  We will clarify it in the revised manuscript.

We fully acknowledge the importance of N2O as a greenhouse gas, particularly for the agricultural sector. However, this study focused on CO2 emissions. Because CO2 emissions from denitrification has never been quantified, and our study demonstrated that it is a significant but overlooked CO2 source. In addition, N2O emissions in groundwater is highly heterogeneous in space and time. Therefore, it is too uncertain to quantify the national budget.

In the revised manuscript, we will explain why we assumed complete denitrification and provide justification of our assumption. |
| L50: what is the ratio of autotrophic and heterotrophic denitrification? | We will provide a summary table of cluster analysis results including the numbers of screens used for the prediction and the predicted area at the cluster level. Organic C-mediated clusters represent heterotrophic denitrification, and pyrite-mediated denitrification represent autotrophic denitrification. |

| L360: When and where is will outgassing to atmosphere? | We addressed this comment above: Further research will be required but it will likely occur when groundwater discharges back to the surface waters. |

---

## Author Response (AR1)

Reviewer 1

General comments
1. This manuscript represents in my opinion a very useful contribution to the multi-faceted research area of denitrification in groundwater systems and will be of substantial interest to a wide audience.
2. While denitrification-related $CO_2$ emissions are featured in the title, the most valuable contributions to the state of the art might possibly lie elsewhere. The (redox) clustering done based on groundwater data from more than 6,000 wells, the cluster interpretation with regard to likely electron donors, and the linkage between clusters and landscape elements, might prove more valuable overall than the estimation of $CO_2$

We revised the title as: A national scale redox clustering for quantifying CO2 emissions from groundwater denitrification.

3. Complete denitrification to $N_2$ is assumed in all calculations presented in this manuscript. The topic of indirect $N_2O$ emissions that could result from incomplete denitrification in groundwater systems (additionally to nitrification) is not mentioned at all. While admittedly not the focus of this study, given the potency of $N_2O$ as GHG, I would like to suggest inserting a brief justification why complete denitrification was assumed, and references to a few studies on indirect $N_2O$ emissions (e.g. by Clough, Weymann, Jahangir, Jurado).

In the revision, the effect of incomplete denitrification was mentioned:

(Line 55-56) *If denitrification is incomplete and terminates at $N_2O$, only 4 moles of DIC are produced per 4 moles of N reduced.*

In addition, we also provided a justification for the complete denitrification assumption:

(Line 89-94*) This study, therefore, aims to quantify $CO_2$ release from denitrification of nitrate derived from agricultural N fertilizer use, in the context of national GHG inventories. To enable this quantification, we assumed complete denitrification. Incomplete denitrification, which produces $N_2O$, is highly heterogeneous in space and time (Clough et al., 2007; Jahangir et al., 2013; Jurado et al., 2017; McAleer et al., 2017). In addition, $N_2O$ produced in groundwater is likely converted to $N_2$, particularly in anoxic groundwater (Jurado et al., 2017). Therefore, we concluded that assuming complete denitrification is a reasonable approximation for large-scale assessments such as this study.*

4. 2a suggests that Cluster 3 is dominant in most of DK, followed by Cluster 7 in the areas not covered by ice sheets during the last glaciation. Pyrite has been identified as the key electron donor in both of these clusters, while organic carbon only appears to serve this role in clusters 1 and 6 (with minor spatial extent), while no clear dominance was evident in clusters 4 and 8. Oxic conditions (Clusters 2 and 5) seem to have insignificant spatial coverage. Given the importance of these findings, I would suggest to explicitly provide information on the spatial extent ($km^2$) of each cluster, the area of pyrite-driven vs. organic carbon-driven denitrification, and references to any field research on electron donors that may underpin these results.

In the revision, we added a table to summarize the redox cluster results and the predictive map.

**Table 1. Summary of cluster analysis and redox cluster prediction**

| Cluster | Number of screens | Predicted area at the redox interface ($km^2$) | Redox stage | Dominant electron donor for denitrification |
|---|---|---|---|---|
| 1 | 246 | 195 | Sulfate-reducing | Organic C |
| 2 | 619 | - | Oxic | No denitrification |
| 3 | 1235 | 26,457 | Fe-reducing | Pyrite |
| 4 | 1940 | 6,027 | Close to sulfate-reducing | Organic C |
| 5 | 221 | - | Oxic | No denitrification |
| 6 | 747 | 743 | Methanogenic | Organic C |
| 7 | 252 | 6,342 | Fe-reducing | Pyrite |
| 8 | 1012 | 3,209 | Fe- and sulfate-reducing to methanogenic | Organic C |

We also added a discussion section to compare our prediction and previous field investigation results:

(Line 329-355) *We compared our predicted redox clusters with findings from previous Danish studies on redox processes at both transect and catchment scales (Jakobsen and Cold, 2007; Jakobsen and Postma, 1999; Kim et al., 2021a, b; Postma et al., 1991; Figure s3). For example, Postma et al. (1991) investigated denitrification along a transect following a groundwater flow path in an unconfined sandy aquifer in Denmark. They found that nitrate concentration decreased rapidly at the redoxcline, primarily by pyrite oxidation, despite the higher abundance of organic matter. This transect corresponds to cluster 7 in our classification, consistent with our prediction.*

*Jakobsen and Postma (1999) also conducted a transect-based field study along a groundwater flow path within the dune area underlain by postglacial sand of central Rømø, Denmark. They*

*investigated how redox processes including iron reduction, sulfate reduction, and methanogenesis vary horizontally and vertically. They concluded that although slow, the fermentation of organic matter controls the co-occurrence of multiple redox processes. Jakobsen and Cold (2007) reported similar findings to Jakobsen and Postma (1999) in an aeolian/post-glacial marine sandy aquifer in northern Zealand. Our redox cluster map indicated that Rømø is primarily predicted as cluster 6 (methanogenesis) and cluster 7 (Fe-reducing), while northern Zealand near the Jakobsen and Cold study site is classified as cluster 4 (close to sulfate reducing), demonstrating reasonable agreement with the earlier transect-based findings. The key role of organic matter fermentation in these study areas further suggests the dominance of organic carbon oxidation as the electron donor for denitrification.*

*At the catchment scale, Kim et al., (2021a) investigated the subsurface structure of denitrification zone in a glacial sediment catchment in Northern Jutland, using a combination of geophysical, geological, hydrological, and geochemical data. By analyzing groundwater chemistry data using K-means clustering analysis, they found that both pyrite oxidation and organic carbon oxidation contribute to denitrification in shallow groundwater. Note that the data used in Kim et al (2021a) were not included in our analysis. They showed that the chemistry of reduced groundwater near the stream showed clear signatures of organic carbon oxidation, while that in the rest of the catchment indicated pyrite oxidation. Consistently, our results identified cluster 4 along the stream, while the remainder of the catchment was predominantly classified as cluster 3 (Fe-reducing, pyrite oxidation; Table 3). Kim et al., (2021b) also carried out a similar study in eastern Jutland in a clay-till catchment. High resolution profiles of groundwater geochemistry revealed that denitrification in this catchment may be primarily driven by pyrite oxidation. This catchment was also predominantly predicted as cluster 3. Overall, our predictions of redox clusters and dominant electron donors for denitrification showed strong agreement with the results of prior process-focused field investigations.*

5. The results suggest that substantial nitrate reduction occurs in most groundwater systems in DK (Fig. 3a). Nevertheless, $CO_2$ emissions attributable to denitrification were estimated to add a maximum of 0.9% to the total emitted $CO_2$ equivalents (see below). While DK has excellent availability of relevant data and scientific expertise, most other countries utilising the IPPC scheme will be less well equipped (and often will have smaller fractions of reduced groundwater). Accordingly, I suggest that most countries are not in a position to credibly estimate what might be a very small contribution relative to all other processes contributing GHG emissions in agricultural landscapes (please

see below for detail). I would like to suggest that resources might be more usefully employed in combatting GHG emissions, rather than in adding small new components to the IPCC accounting system. Please consider these points when revising your Conclusions.

In the revision, we explained that the IPCC guidelines' requirement for individual accounting for each greenhouse gas to justify why it is important to quantify $CO_2$ emissions from denitrification:

(Line 85-88) *Compared to methane ($CH_4$) and $N_2O$, $CO_2$ contributes a minor share of the total GHG emissions from agriculture. However, the IPCC guidelines require individual accounting for each GHG unless there are specific methodological reasons for aggregation (IPCC, 2006). Thus, all anthropogenic sources of $CO_2$ in agriculture are required to be accounted for, regardless of magnitude*

We revised our discussion on the importance of $CO_2$ emissions from denitrification in a more measured tone and added a possibility of validation through existing research:

(Section 3.4, Line 425-433) *Our results indicate that groundwater denitrification may represent a significant anthropogenic source of $CO_2$—comparable in magnitude to liming and substantially larger than other $CO_2$ sources currently included in the IPCC guidelines. These findings imply that current estimations of $CO_2$ emissions from the agricultural sector may be underestimated. While further evaluation is needed, our findings suggest that $CO_2$ emissions from denitrification should be considered in future revision of the IPCC GHG inventory guidelines. These results would benefit from validation through additional studies across diverse settings. Denitrification is one of the most extensively investigated biogeochemical processes globally, and findings from these studies may help estimate $CO_2$ emissions from denitrification in groundwater, and potentially in streams and estuaries under varying agricultural, climatic and geological conditions. By synthesizing existing research, $CO_2$ emission factors for denitrification could be more accurately constrained at both local and national levels.*

(Conclusion, Line 454-458) *These findings suggest that the current $CO_2$ emissions from the agriculture sector are likely underestimated, and that subsurface denitrification may be a non-negligible component. While $CO_2$ is a relatively minor component of the overall agricultural GHG budget, our findings highlight that groundwater denitrification represents a previously*

*unaccounted anthropogenic $CO_2$ source. We recommend that this process should be considered in future efforts to improve the completeness of agricultural $CO_2$ inventories.*

We also added a section in Conclusion to mention the importance of multi-disciplinary research in upscaling process-based knowledge up to larger scale quantifications:

(Line 459-467) *This study also highlights the value of integrating process-based understanding with data-driven methods to address the challenges posed by spatial heterogeneity and the upscaling of complex subsurface biogeochemical processes such as greenhouse gas emissions. While the mechanisms and primary controls of denitrification have been extensively studied at the small spatial scales such as profiles, transects, and catchment, translating this knowledge into robust, large-scale quantification has remained challenging. By integrating insights from hydrogeology, groundwater redox chemistry, and long-term monitoring data within a predictive mapping framework, we demonstrated how multidisciplinary approaches—including machine learning—can integrate fundamental process-based understanding. This integrative approach offers a promising pathway not only for improving nitrate management strategies but also for reducing uncertainties in greenhouse gas inventories from agricultural systems and more generally, for large scale studies on groundwater geochemistry.*

| 6. The Specific comments listed below are largely of a minor or technical nature, but addressing them should improve the clarity of the manuscript. | |
| --- | --- |
| 15: what is meant by 'dominant denitrification processes'? Different electron donors driving denitrification? | Revised as follows (Line 15-17) *A set of machine learning techniques was applied to cluster groundwater redox conditions and map the dominant electron donors for denitrification at the national scale.* |
| Table 1: Equations 1 and 2 are both assuming complete denitrification to $N_2$. Could you please add a sentence on the effect incomplete denitrification would have. | Revised as follows (Line 56-57) *If denitrification is incomplete and terminates at $N_2O$, only 4 moles of DIC are produced per 4 moles of N reduced.* |
| 65 ff: The calculations marked by * and ** in Table 1 are valid for situations where calcite saturation occurs. Could you please provide | It will depend on the underlying geology, particularly the content of calcite. Therefore, it is difficult to generalize it. However except for |

| | |
|---|---|
| the reader with information on how common such conditions are within the groundwater system and where groundwater discharges into surface water bodies? Could it be argued that the $CO_2$ emissions estimates represent an upper limit? | Western Jutland the aquifers contain calcite from near the groundwater table. Even in Western Jutland much of the water will flow through calcite-bearing layers before reaching surface waters. So yes, it is an upper limit of $CO_2$ emissions for pyrite oxidation-denitrification process, in most settings the water will be equilibrated with calcite. |
| 68: 'triggered by anthropogenic nitrate input'. Not all N in groundwater originates from fertiliser application. Is the fraction of the denitrified N that might have come from natural sources considered negligible? | Yes, other sources of N such as natural sources of N or wastewater were considered insignificant. |
| 92: 'map of denitrification processes' seems a misnomer. The map is showing the distribution of six clusters with reduced groundwater redox chemistry. | In the revision, we revised as follows: (Line 100-101) *prediction of a national map of denitrification electron donors;* |
| 93: Please make sure you clearly define in Section 3.4 what exactly you mean by 'agriculture GHG inventory'. | Revised as follows: (Line 101-102) *quantification of the $CO_2$ emissions from groundwater denitrification in the context of the agricultural GHG emissions in Denmark.* |
| 106: Given that at least five measurements were required over the entire period (1890-2022), can you please provide the reader with a summary statement from which period most of the used data originate (e.g. 80% of the data were collected between 2001 and 2022)? Can we assume that the analysis is not affected by concentration trends during this period? | The final data that were used to the cluster analysis were primarily from 1990-2020.

Line 113: ..., *primarily collected between 1990-2020.*
The concentrations change over time, which might explain the variability (or wide ranges) of some of the constituents that we included in the analysis. However, because we were interested in the stoichiometric ratios of products of denitrification reactions, the effects of the temporal trends of groundwater chemistry assuming the dominant reactions are not changing will be minor. If dominant reactions are changing it will result in less clear clustering. |
| Sections 2.2 and 2.3: I would like to disclose that my understanding of ML techniques is | We wanted to mention that the ML methods employed in here is widely used in similar |

| | |
|---|---|
| very limited, and therefore cannot evaluate the choice of methods. Another reviewer may be able to fill this gap. | applications. In addition, both Matlab and python codes for these analyses are readily available. |
| 107: Numbers are reported for 'screens' rather than bore/well sites. Does this account for multiple screens possibly being located at different depths at one site? | Revised as follows:

Line 108: *Some wells have multiple screens.* |
| 112: 'The cleaned dataset was analyzed to categorize redox conditions and to identify dominant processes by combining two machine learning techniques'. 1) Does 'dominant processes' refer to nitrate reduction processes (e.g. driven by pyrite vs organic carbon)?; 2) Before embarking on ML techniques, have you tried to characterise the redox conditions using the 'classical' framework by McMahon & Chapelle (2008)? | 1) We first identified redox conditions and processes that are responsible for the cluster's chemical signature, not only denitrification process. For instance, cluster 6 was characterized by high methane and we identified methanogenesis may be responsible for this cluster's chemistry.
2) No, we did not use pre-defined redox conditions. Our approach was to employ data-driven techniques, i.e., MNF and K-means clustering to identify different redox conditions. |
| 134: The oxic clusters 2 and 5 are shown in Fig. s2, not Fig. s1 as stated. | Corrected to Figures s2. |
| 158: 'the redox interface' is defined as 'the bottom of the nitrate-reducing zone'. Maybe specify 'the first redox interface', as Koch et al. (2024) makes it clear that more complex vertical stratification occurs widespread in DK. | Revised as follows:

Line 164-168: *In Denmark, due to glaciotectonic deformation during the most recent glaciations, the complexity of the redox architecture varies significantly, resulting in multiple redox interfaces (Kim et al., 2019; Koch et al., 2024). Koch et al. (2024) predicted the depth to the first redox interface as well as its structural complexity at the national scale at 25m x 25m resolution based on sediment color data and 20 explanatory variables (Table 2) using a gradient boosting with decision tree (GBDT) algorithm (Koch et al., 2024).* |
| 169/170: Could you please provide the absolute number or percentage of screens excluded? | Revised as follows:
Line 177-176: For the prediction of the redox cluster map, we first excluded groundwater |

| | screens from 1) oxic clusters (about 14% of the total screens); and 2) reduced clusters with a Silhouette score less than 0 (about 13 % of the total screens).

In addition, Table 3 shows the summary of cluster analysis and map prediction results. |
|---|---|
| 172: 'depths of groundwater screens shallower than the depths of redox interface minus 5 (D5), 10 (D10), and 15 (D15) meters' is unclear; please reformulate this explanation. The caption to Fig. s1 suggests that e.g. D5 stands for wells with 'screen tops deeper than 5 (…) meters below the redox interface'. However, the corresponding well numbers given for D5 (235), D10 (566), and D15 (1019) seem to contradict this information. Does D5 stand for all wells where the screen is a maximum of 5m below the redox interface? | Revised as follows:
Line 181-182: *those where the depths of screen top was no more than 5 (D5), 10 (D10), and 15 (D15) meters below the redox interface.* |
| 174: First time 'wells' is used rather than 'screens'. Maybe consider using one term throughout the manuscript or clarify why different terms are used if there is a reason for it. | Corrected to 'screens'. |
| 187: The 1990-2010 period was used for nitrate reduction estimates. Were the measurements from the 6,273 screens (line 108) also predominantly from this period? | It is predominantly from 1990 to 2020. Most data are from 1990-2010, which is synchronized with the nitrate reduction estimation. |
| 208 ff: Could you please provide the number of wells in each of the eight identified clusters. Would it be useful to apply the USGS redox classification scheme to the wells in these clusters? Also, could the clusters interpreted as reflecting heterotrophic denitrification be grouped (and presented) according to the redox sequence (weakly to strongly reduced: 2,5<4<8<1<6)? | Revised as follows:

Line 217-220: *Our results showed that the Danish groundwater can be categorized into eight clusters): two oxic clusters (cluster 2 and 5) and six reduced clusters (1, 3, 4, 6, 7, and 8), each at various redox stages (Figure 1c). Cluster 4 was the most frequent cluster (1940 screen), followed by cluster 3 (1235 screens), cluster 8 (1012 screens), cluster 6 (747),* |

| | |
|---|---|
| | *cluster 2 (619 screens), cluster 7 (252 screens), cluster 5 (221 screens), and cluster 1 (246; Table 3).*

Line 272-274: *Altogether, the redox sequence of the clusters can be summarized according to the redox ladder: cluster 2 and 5 as oxic, cluster 3 and 7 as Fe-reducing, cluster 4 and 8 as transitioning from Fe-reducing to sulfate reducing, cluster 1 as sulfate-reducing, and cluster 6 as methanogenic (Table 3).* |
| Please also consider if the key cluster information provided in Sections 3.1 and 3.2 could usefully be presented in a Table? This would facilitate direct comparison between clusters, both concerning their chemistry and spatial distribution. | We added a table (Table 3) for summarizing the results of cluster analysis and map prediction. |
| Given the variability in the data within a cluster (e.g. Fig. 1c), could some variability be interpreted as indicating that nitrate may have been reduced along its flowpath to the well screen by a combination of heterotrophic and autotrophic denitrification? | Yes, that is possible. Although we identified one dominant denitrification process for each cluster, it is absolutely possible that different denitrification processes occur along the pathways. In addition, groundwater mixing and variation in time can contribute to the variability. Our interpretations of dominant denitrification process, however, was based on not only groundwater chemistry but also hydrogeological features; thus, it enabled us to identify the most probable process for denitrification. |
| 270 ff: It would seem useful to start here with info on the spatial extent (km$^2$ or % of DK area) of the clusters, as Cluster 3 appears to be dominant, followed by Cluster 7, and all others well behind. Accordingly, pyrite would appear to provide much more widespread denitrification potential in DK than organic carbon. | Table 3 shows the area of each cluster predicted in our study. |
| 273: Final 'maps of denitrification processes' (Fig. 2a). I find the use of the term 'process' | It is revised as follows: |

| | |
|---|---|
| somewhat misleading. As I understand it, Fig. 2a represents a spatial prediction of groundwater chemistry clusters. As outlined in Section 3.1, these clusters are thought to reflect the prevalence of one or more of the reactions listed in Table 1. Accordingly, I would suggest replacing 'denitrification processes' with 'denitrification clusters' or even wider 'redox clusters' (as denitrification reactions are only a subset of the reactions defining the clusters). | Line 287-288: *The final maps (Figure 2a and Supplementary Figure s1) of redox clusters were generated by GBDT models trained on 100% of the available data.*

And for the rest of the manuscript, "map of denitrification processes" was revised to "map of redox clusters". |
| Fig. 2a: Maybe move the label 'Main stationary line' out of the black square that indicates the enlarged area, to make it clear that it refers to the somewhat inconspicuous dotted line, not the more prominent square. I also wonder, how to better present the less prominent clusters? Maybe colours could be swapped between Clusters 4 and 6, so that Cluster 4 areas in the still fairly small enlargement can be more easily recognised? Making Cluster 4 more prominent would also help with the discussion of Fig. 3 (highest DIC production in northern Jutland). | 'Main stationary line' was moved outside the box. We tried different color combinations, but this was the best one to represent all the clusters. The area of cluster 1 was too small (only 0.5% of the total area); therefore, it is difficult to present. Cluster 6 and 8 were the next smallest clusters. These bright colors display them well, we believe. |
| 280/81: Maybe replace 'outside' with 'west and south' and 'behind' with 'east and north'? | Revised as suggested:

Line 294-296: *In contrast, cluster 3, interpreted similarly but under carbonate-rich conditions, was mainly found east and north of the main stationary line.* |
| 315ff: Please either add 'Jutland' and 'Zealand' labels on the map or provide more location info in the text (e.g. in the west of DK). | Revised as suggested. |
| 324 ff: I would suggest emphasizing more that the spatial patterns of nitrate reduction and DIC production differ substantially, as the electron donors fuelling denitrification differ spatially. | Revised as follows:
Line 370-376:

*Although western Jutland showed the highest nitrate reduction in groundwater, DIC production was moderately high, ranging from* |

| | |
|---|---|
| | *60 to 145 kg $CO_2$ ha$^{-1}$ yr$^{-1}$. While the highest DIC production was predicted in northern Jutland (up to 180 kg $CO_2$ ha$^{-1}$ yr$^{-1}$; Figure 3b) despite the low to moderate nitrate reduction in groundwater (< 60 kt of N yr$^{-1}$). Such results can be attributed to differences in dominant electron donors. In the northern Jutland, denitrification was predicted to be mediated by organic C (Figure 3a). Compared to pyrite oxidation, which releases 0.33 moles of DIC per mole of nitrate reduction, organic matter-mediated denitrification increases 1.25 moles of DIC per mole of N reduction (Table 1). Consequently, more DIC, thus more $CO_2$, is produced by denitrification in this region.* |
| Section 3.4: Notwithstanding the GHG contributions by LULUCF, the 'agricultural contributions' in the narrow sense comprise $CH_4$, $N_2O$, and $CO_2$ from liming, urea, and other fertilisers. It would seem to me that the 90-104 kt estimated below almost pale into insignificance relative to the total GHG emissions attributed to 'agriculture' (amounting to 11,268 kt $CO_2$-eq. yr$^{-1}$, see Fig. 4). | We addressed this comment above (see major comment 5). We added the IPCC guidelines' requirement for individual accounting for each greenhouse gas. |
| 338: I'm unsure if 'excluding' is the right word here? Would 'after' be more suited? | It is "excluding" – meaning without contributions from the total national GHG inventories. |
| 375: If I understand the numbers correctly, the upper limit of 104 kt (Fig. 4) would result in an increase of $CO_2$ equivalents of 0.9%; the $CO_2$ contribution to GHG emissions rising from 2.3 to 3.1% (358 out of 11,372 kt). While acknowledging that substantially smaller contributions are accounted for in the IPCC guidelines, these are more easily quantifiable (e.g. from fertiliser sales statistics). I am unconvinced that estimating $CO_2$ resulting from denitrification could be added to the IPPC procedure in a credible manner. DK may | We have addressed this comment above (see major comment 5). |

| | |
|---|---|
| be in the enviable position of being a virtual laboratory, but even under the favourable Danish conditions the estimates rely on a number of assumptions which introduce uncertainty. Estimates for most other countries around the world would inevitably be markedly less certain than the results presented here. | |
| 405: I find the 38% number for 'agricultural emissions' misleading, as the 254 kt calculation basis refers in the IPPC system only to the minor contributions made by liming, urea and other fertilisers (254 kt), rather than the total of 11,268 kt $CO_2$-eq. attributed to agriculture (incl. 5132 kt arising from $N_2O$ and the 5881 kt from $CH_4$, Fig. 4). | We deleted this part and rephrased our conclusion with a more conservative tone (major comment 5). |

Reviewer 2

The manuscript "Nitrate reduction in groundwater as an overlooked source of agricultural $CO_2$ emissions" provided an estimation of $CO_2$ emissions by heterotrophic Denitrification from groundwater, based on monitoring data of Danish groundwater and come up with the hypothesis that heterotrophic Denitrification is a significant source for DIC and outgassing $CO_2$ and should be taken into account for the national GHG emissions estimations and that this may also important for other countries.

Generally, to conduct an entire GHG budget the emissions from groundwater have to be taken into account and a robust estimation is necessary, nevertheless some questions arise.

- At which point and time the exchange of the GHG including $CO_2$ between groundwater and atmosphere happened. After the leaching to river systems and marine waters, or at a earlier point. So maybe to $CO_2$ from submarine groundwater discharge (SGD) already count for the general GHG budget

In the revision, we added the following sentences:

Line 397-399: *As groundwater discharges back into surface waters such as streams, $CO_2$ will be degassed from groundwater because groundwater is oversaturated with $CO_2$ with respect to the atmosphere.*

We acknowledge that in Denmark, we have not quantified the contribution of submarine groundwater discharge (SGD) to the national water budget. While SGD might be an important contributor at a local scale water and nutrient budgets, its contribution at larger scales will be minor. Therefore, we concluded that SGD is a minor pathway to release CO2 emissions from denitrification in groundwater.

- You just mentioned heterotrophic denitrification, because that produced $CO_2$. But what is the percentage of autotrophic denitrification in the systems and does that play a role for $CO_2$ fixation?

In this study, we considered both heterotrophic (i.e., organic carbon mediated) and autotrophic (i.e., pyrite-oxidation mediated) denitrification. These two processes have been identified as the dominant denitrification reactions in Danish groundwater as well as in other regions with similar geological setting.

In case of autotrophic denitrification by pyrite oxidation, the process contributes to an increase of DIC. This occurs because pyrite oxidation generates protons, which promotes calcite dissolution if calcite is present. While some other form of autotrophic denitrification processes

that fix $CO_2$ (such as driven by oxidation of sulfur S(0) or Fe(II) occur under reduced conditions. Our analysis showed that these reactions may be limited to a spatial extent to the redox interface. For instance, Cluster 1 (sulfur-reducing conditions) and 6 (methanogenesis) were predicted to account for only 0.5 % and 2% of the total area at the redox interface. We interpreted that these conditions typically occur in organic-rich environments, where organic-mediated denitrification would likely dominate and reduce nitrate at shallower depth before groundwater reaches these deeper, more reduced zones.

In the revised manuscript, we provided a table (Table 3) summarizing the results of the cluster analysis and map predictions.

> - Anaerobic denitrification also produced TA, how that was taken into account and maybe increase the capability to store DIC and emit less. See: Middelburg, J. J., Soetaert, K., and Hagens, M.: Ocean Alkalinity, Buffering and Biogeochemical Processes, Reviews of Geophysics, 58, e2019RG000681, https://doi.org/10.1029/2019RG000681, 2020.

Yes, both heterotrophic and autotrophic denitrification reactions considered in this study contribute to an increase in dissolved inorganic carbon in groundwater. The increase in DIC also elevates $pCO_2$ in groundwater, resulting in $CO_2$ degassing when the supersaturated groundwater encounters the atmosphere. Therefore, while denitrification does temporally increase DIC in groundwater, this acts as a short-term C storage, typically on the order of years to decades, before the $CO_2$ is eventually released into the atmosphere. It is also important to highlight that denitrification in groundwater mineralize and mobilize both organic and inorganic C pools that would otherwise remain stored over a geological time scale.

The suggested reference, on the other hand, focuses on total alkalinity in ocean systems, where the residence times of both water and carbon are significantly longer than in groundwater systems. This extended residence times allows for long-term carbon storage and buffering in the ocean, which contrast with the more dynamic and short-term nature of the C cycle in groundwater.  Therefore, while the oceanic context provides valuable insights into global carbon cycling and buffering, its direct comparison to groundwater systems may not be fully applicable. Therefore, we will not implement any changes to the revised manuscript.

Some specific comments:

| | |
|---|---|
| L 18/19: Why you mention $CO_2$-eq. and not just $CO_2$ although is it as DIC. Where and when the 50% emitted to the atmosphere | It is displayed as $CO_2$-eq because this number can be compared to the other emissions given as $CO_2$-eq emissions. Therefore, we will keep $CO_2$-eq as the unit to express DIC. The second question was addressed above. |
| L 35/41: this paragraph raises some questions. Nitrogen fertilizers are more than nitrate, so that also organic nitrogen and ammonium is part of that. So in consequence nitrification plays also a crucial role and can be a significant source of N2O. The references for the N2O sources Ritchie at al, 2023 just focused on "anthropogenic" Sectors. So that natural processes and sources with is maybe also anthropogenic impacted are not negligible. Especially ODZ and also groundwater discharge can be source of N2O and other GHGs | In the revised manuscript it was changed to "*nitrogen fertilizers and manure*" (Line 36 and Line 42).

 We fully acknowledge the importance of N2O as a greenhouse gas, particularly for the agricultural sector. However, this study focused on $CO_2$ emissions. Because $CO_2$ emissions from denitrification has never been quantified, our study demonstrated that it is a significant but overlooked $CO_2$ source. In addition, N2O emissions from groundwater is highly heterogeneous in space and time. Therefore, it is too uncertain to quantify the national budget.

 In the revised manuscript, we explained why complete denitrification was assumed:

 Line 90-94: *To enable this quantification, we assumed complete denitrification. Incomplete denitrification, which produces $N_2O$, is highly heterogeneous in space and time (Clough et al., 2007; Jahangir et al., 2013; Jurado et al., 2017; McAleer et al., 2017). In addition, $N_2O$ produced in groundwater is likely converted to $N_2$, particularly in anoxic groundwater (Jurado et al., 2017). Therefore, we concluded that assuming complete denitrification is a reasonable approximation for large-scale assessments such as this study.* |
| L50: what is the ratio of autotrophic and heterotrophic denitrification? | Table 3 summarizes the results of cluster analysis and map prediction. In addition, in the abstract we specified as follows:
 Line 17-18: *At the redox interface, denitrification was predicted to be mediated* |

| | *by pyrite oxidation in approximately 76% of the area with the remainder dominated by OC oxidation.* |
|---|---|
| L360: When and where is will outgassing to atmosphere? | We added a sentence as follows:

Line 397-399: *As groundwater discharges back into surface waters such as streams, $CO_2$ will be degassed from groundwater because groundwater is oversaturated with $CO_2$ with respect to the atmosphere.* |

---

## Author Response (AR2)

Reviewer 1

Suggestions for revision or reasons for rejection
(visible to the public if the article is accepted and published)

The authors are to be congratulated on their thorough and comprehensive response to the comments previously provided, which improved the manuscript significantly.

In my opinion, the only remaining weak point concerns the authors' recommendation to incorporate denitrification-induced CO2 emissions into the IPCC framework. My reasoning is as follows:

Authors: 'In addition, we also provided a justification for the complete denitrification assumption: (Line 89-94) This study, therefore, aims to quantify CO2 release from denitrification of nitrate derived from agricultural N fertilizer use, in the context of national GHG inventories. To enable this quantification, we assumed complete denitrification.'

However, to the best of my understanding, IPCC guidelines for national GHG inventories assume indirect N2O emissions to occur from nitrate leached from agricultural soils (EF5). My concern is that the current IPCC methodology using the EF5 factor (implying some degree of incomplete denitrification) would be incompatible with introducing an additional new approach for accounting for denitrification-induced CO2 emissions (assuming complete denitrification). Do you agree that one of the two options would have to be chosen?

The reviewer has a good point. Indeed, assuming complete denitrification would theoretically preclude N2O emissions. However, N2O emissions from groundwater represent only a small fraction of the total groundwater N retention. While the fraction may vary considerably across sites, global estimates suggest that only 1 % of total groundwater denitrification is emitted as N2O (Bouwman et al., 2013). Similarly, the IPCC guidelines' emission factor for indirect N2O via groundwater is 0.6% of leached N. N2O is a potent greenhouse gas due to its high global warming potential, not because of large quantities emitted from groundwater.

Therefore, we believe that N2O and denitrification-induced CO2 emissions should be treated separately with different level of uncertainties. For CO2 emissions from denitrification, a 1 % of uncertainty is acceptable, where as N2O emissions require different types and levels of data and modeling approaches.

In the revised manuscript, we have strengthened our argument for the complete denitrification assumption:

Line 89-99:

*This study, therefore, aims to quantify $CO_2$ release from denitrification of nitrate derived from agricultural N fertilizer use, in the context of national GHG inventories. To enable this quantification, we assumed complete denitrification. While incomplete denitrification can result in the production of N2O, which has a substantial climate impact, this $N_2O$ is typically*

*further reduced to $N_2$ in deeper anoxic aquifer (Jurado et al., 2017). As a result, the fraction of $N_2O$ emitted from groundwater is generally insignificant compared to the total amount of nitrate denitrified in groundwater. For instance, Bouwman et al. (2013) estimated that about 1 % of total groundwater denitrification results in $N_2O$ emissions, and the most recent IPCC guidelines suggest that 0.6% of leached N would be ultimately emitted as $N_2O$ from groundwater (IPCC, 2019; Tian et al., 2019). Furthermore, $N_2O$ production and reduction processes are highly heterogeneous in space and time (Clough et al., 2007; Jahangir et al., 2013; Jurado et al., 2017; McAleer et al., 2017). While these hotspots and hot moments may be relevant for local-scale assessment, they are unlikely to significantly influence large-scale GHG budgets. Therefore, we consider that assuming complete denitrification is a reasonable approximation for large-scale assessments such as this study.*
* * *
Authors: 'Incomplete denitrification, which produces N2O, is highly heterogeneous in space and time (Clough et al., 2007; Jahangir et al., 2013; Jurado et al., 2017; McAleer et al., 2017).'

This is correct, but per se not a justification for ignoring it. Despite its immense small-scale variability (hot spots, hot moments), incomplete denitrification could still be quantitatively very important in terms of GHG potential, especially when integrated over larger areas and longer time scales.
* * *
We disagree with the reviewer's point. In our view, hotspots and hot moments may influence short-term, local-scale budgets, but they are unlikely to have a significant impact on long-term, large-scale greenhouse gas budgets. Over larger spatial and temporal scales, these localized and transient events tend to average out, and the overall budget converges toward the system-wide mean. We have addressed this reasoning in our response to the previous comment.
* * *
Authors: 'In addition, N2O produced in groundwater is likely converted to N2, particularly in anoxic groundwater (Jurado et al., 2017). Therefore, we concluded that assuming complete denitrification is a reasonable approximation for large-scale assessments such as this study.'

In many situations, groundwater denitrification will indeed almost quantitatively proceed to N2 gas. However, given that N2O is a GHG 298 times more potent than CO2, I would like to encourage you to provide the reader with one or more example calculations that put your estimated complete denitrification-induced CO2 emissions into perspective. E.g., how would the GHG effect of currently unaccounted for CO2 emissions compare to that of N2O emissions if only 90% of the nitrate was fully reduced to N2 (and 10% to N2O)?

As mentioned earlier, the fraction of N2O emission from groundwater is generally estimated to be around 1% or less of the total groundwater denitrification. However, to put denitrification induced CO2 emissions into perspective, we have added the following sentence to the revised manuscript:

Line 426-427:

*For comparison, the national GHG inventories of Denmark estimates that 360 kt of $CO_2$-eq $yr^{-1}$ of $N_2O$ would be emitted via groundwater in 2022 (Nielsen et al., 2024).*

We also identified more recent national GHG inventory data and have updated figure 4 accordingly to reflect these new estimates.

[Figure]

L 448: Please change 205 kt to 204 kt.

Corrected to 204.

Reviewer 2

Suggestions for revision or reasons for rejection
(visible to the public if the article is accepted and published)
The authors of the manuscript with the new title "A national scale redox clustering for quantifying CO2 emissions from groundwater denitrification" put a lot of efforts in the revision of the paper according to the suggestions of the editor and the two reviewers. They have taken up the criticism and provided necessary explanations for some assumptions, such as the relationship between autotrophic and heterotrophic denitrification or the role of N2O formation during incomplete denitrification. I can

understand the explanations and find the considerations a good addition to the GHG emission estimates.

Only one smaller issue I want to rise, is the relation to TA by anaerobic metabolisms. Sure, the suggested reference is about the situation in the Ocean, but that doesn't prove that it only plays a role in the ocean. You should at least discuss that.

Thank you for the comment. The process the reviewer refers to i.e., denitrification linked to anaerobic metabolism is already represented in our study. Denitrification coupled with organic carbon oxidation as analyzed in our study included the alkalinity effects associated with anaerobic heterotrophic metabolism. Therefore, we believe this process is already addressed in our analysis, and no changes have been made in response to this comment.